# BLIND INVERSION USING LATENT DIFFUSION PRIORS

## ABSTRACT

Diffusion models have emerged as powerful tools for solving inverse problems due to their exceptional ability to model complex prior distributions. However, existing methods predominantly assume known forward operators (*i.e.*, non-blind), limiting their applicability in practical settings where acquiring such operators is costly. Additionally, many current approaches rely on pixel-space diffusion models, leaving the potential of more powerful latent diffusion models (LDMs) underexplored. In this paper, we introduce LatentDEM, an innovative technique that addresses more challenging blind inverse problems using latent diffusion priors. At the core of our method is solving blind inverse problems within an iterative Expectation-Maximization (EM) framework: (1) the E-step recovers clean images from corrupted observations using LDM priors and a known forward model, and (2) the M-step estimates the forward operator based on the recovered images. Additionally, we propose two novel optimization techniques tailored for LDM priors and EM frameworks, yielding more accurate and efficient blind inversion results. As a general framework, LatentDEM supports both linear and non-linear inverse problems. Beyond common 2D image restoration tasks, it enables new capabilities in non-linear 3D inverse rendering problems. We validate LatentDEM's performance on representative 2D blind deblurring and 3D pose-free sparse-view reconstruction tasks, demonstrating its superior efficacy over prior arts.

## 1 INTRODUCTION

Inverse problems aim to recover underlying signals $x$ from partial or corrupted observations $y$ generated by a forward operator $\mathcal{A}_\phi(\cdot)$. Such problems are prevalent in computer vision and graphics, encompassing a variety of tasks ranging from 2D image restoration(denoising, deblurring, and inpainting (Bertero et al., 2021; Bertalmio et al., 2000)) to 3D reconstruction(CT, NLOS, inverse rendering (Marschner, 1998; Mait et al., 2018; Faccio et al., 2020)), *etc*. Typically, inverse problem solvers assume the forward model $\mathcal{A}$ and its physical parameters $\phi$ are known (*i.e.*, *non-blind*) (Schuler et al., 2013). However, acquiring accurate forward models is often challenging or impractical in real-world settings. This necessitates solving *blind* inverse problems, where both the hidden signals $x$ and the forward model parameters $\phi$ must be jointly estimated.

Being heavily ill-posed, inverse problems largely rely on data priors in their computation. Traditional supervised learning approaches train an end-to-end neural network to map observations directly to hidden images ($y \rightarrow x$) (Li et al., 2020; Jin et al., 2017; McCann et al., 2017). Recently, diffusion models (DMs) (Ho et al., 2020; Song et al., 2020; Sohl-Dickstein et al., 2015) have emerged as powerful inverse problem solvers due to their exceptional ability to model the complex data distribution $p(x)$ of underlying signals $x$. DMs approximate $p(x)$ by learning the distribution's score function $\nabla_{x_t} \log p_t(x_t)$ (Song & Ermon, 2019), allowing data-driven priors to be integrated into Bayesian inverse problem solvers (*e.g.*, diffusion posterior sampling (Chung et al., 2022b)). Later, latent diffusion models (LDMs) have evolved as a new foundational model standard (Rombach et al., 2022) by projecting signals into a lower-dimensional latent space $z$ and performing diffusion there. This strategy mitigates the curse of dimensionality typical in pixel-space DMs and demonstrates superior capability, flexibility, and efficiency in modeling complex, high-dimensional distributions, such as those of videos, audio, and 3D objects (Rombach et al., 2022; Wang et al., 2023; Stan et al., 2023; Blattmann et al., 2023).

Although both DM-based and LDM-based solvers have demonstrated impressive posterior sampling performance in diverse computational imaging inverse problems, existing methods predominantly fo-

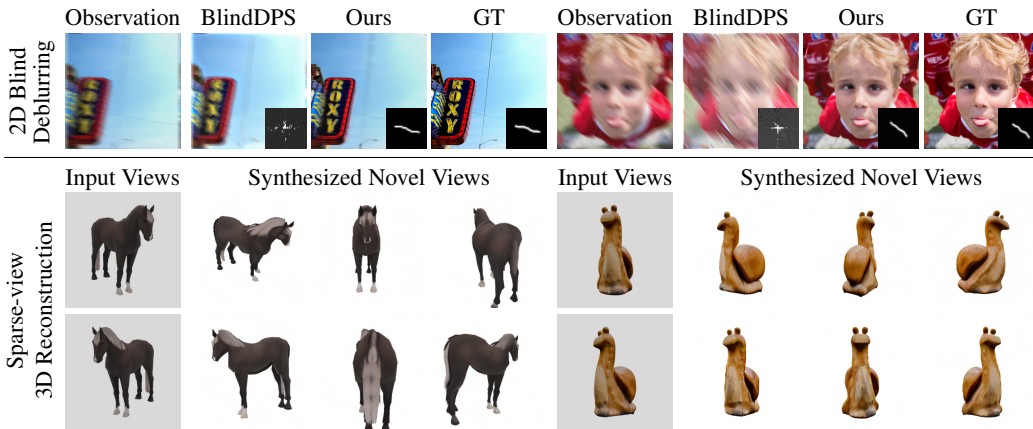

Figure 1: We apply our method on two representative blind inverse problems: **2D Blind Deblurring** and **Pose-free Spare-view 3D Reconsturction**. Notably, in 2D task, our method achieves more accurate image recovery and kernel estimation over BlindDPS Chung et al. (2022a), while in 3D task we successfully reconstruct consistent novel view images from unposed input views.

cus on non-blind settings (*i.e.*, optimizing images $x$ with known forward model parameters $\phi$) (Chung et al., 2022b; Rout et al., 2024; Song et al., 2023). Blind inversion poses more challenges since jointly solving $x$ and $\phi$ involves non-convex optimization, often leading to instabilities. While recent advances have explored the feasibility of solving blind inverse problems using pixel-based DMs (Laroche et al., 2024; Chung et al., 2022a), these methods suffer from computational inefficiencies and limited capability in modeling complex image priors, rendering them unsuitable for more challenging, high-dimensional blind inversion tasks like 3D inverse rendering.

In this paper, we introduce LatentDEM, a novel approach that solves blind inverse problems using powerful LDM priors. The core concept of LatentDEM involves a variational EM framework that alternates between reconstructing underlying images $x$ through latent diffusion posterior sampling (E-step) and estimating forward model parameters $\phi$ using the reconstructed images (M-step). We further design an annealing optimization strategy to enhance the stability of the vulnerable latent space optimization, as well as a skip-gradient method to accelerate the training process. Consequently, LatentDEM allows us to leverage the capabilities of pre-trained foundational diffusion models to effectively solve a wide range of blind 2D and 3D inverse problems.

To the best of our knowledge, LatentDEM is the first method that incorporates powerful LDM priors (Rombach et al., 2022) in the blind inverse problems. We first validate our method with Stable Diffusion (Rombach et al., 2022) priors and perform the representative 2D blind motion deblurring task, where we showcase superior imaging quality and efficiency over prior arts. LatentDEM further demonstrates new capabilities in more challenging non-linear 3D inverse rendering problems. Given a set of unposed sparse-view input images, we apply Zero123 priors (Liu et al., 2023b) to synthesize the corresponding novel view images, supporting pose-free, sparse-view 3D reconstruction. Our results exhibit more 3D view consistency and achieve new state-of-the-art novel view synthesis performance.

## 2 RELATED WORK

**Inverse Problems.** The goal of general inverse problems is to recover signals $x \in \mathbb{R}^D$ from partial observations $y \in \mathbb{R}^M$:

$$y = \mathcal{A}_\phi(x) + n, \tag{1}$$

where $\mathcal{A}$, $\phi$ and $n \sim \mathcal{N}(0, \sigma^2 \mathbf{I})$ represent the forward operator, its parameters, and the observation noise, respectively. The signal $x$ can be either solved by supervised learning approaches (Li et al., 2020; Jin et al., 2017; McCann et al., 2017), or recovered within the Bayesian framework to maximize the posterior: $p(x|y) \propto p(x)p(y|x)$, where data priors $p(x)$ are of vital importance. Traditional methods use handcrafted priors such as sparsity or total variation (TV) (Kuramochi et al., 2018; Bouman & Sauer, 1993). However, these priors cannot capture the complex natural image distributions, limiting the solvers' ability to produce high-quality reconstructions (Danielyan et al.,

2011; Ulyanov et al., 2018; Candes & Romberg, 2007). Recent advances in diffusion models (DMs), particularly latent diffusion models (LDMs), have made them attractive for inverse problems due to their powerful data prior modeling capabilities (Chung et al., 2022b; Rout et al., 2024; Song et al., 2023). In this paper, we focus on solving *blind* inverse problems using latent diffusion models (Liu et al., 2023b; Rombach et al., 2022).

**Diffusion Models for 2D Inverse Problems.** DMs have been applied to a wide range of 2D inverse problems, including natural image deblurring, denoising, super-resolution and fusion tasks (Wang et al., 2022; Chung et al., 2022b;c;d; Feng & Bouman, 2023; Zhao et al., 2023), as well as medical and astronomy image enhancement (Song et al., 2021a; Chung & Ye, 2022; Wang et al., 2022). Diffusion Posterior Sampling (DPS) pioneered the use of DMs as strong data priors to solve non-blind 2D inverse problems in a maximum-a-posteriori (MAP) manner (Chung et al., 2022b;c;d). Later works (Rout et al., 2024; Song et al., 2023) evolved DPS with Latent Diffusion Model (LDM) priors, demonstrating improved performance due to better priors. While these methods (Chung et al., 2022b; Rout et al., 2024; Song et al., 2023) all address non-blind problems, BlindDPS (Chung et al., 2022a) extends DPS to the blind setting by modeling diffusion priors of both data and forward model parameters. Similar to our approach, FastEM (Laroche et al., 2024) proposes to address blind inversion within an Expectation-Maximization (EM) framework. However, Chung et al. (2022a); Laroche et al. (2024) remain limited to pixel-based DMs, as the instability of LDMs makes the optimization even harder. In this paper, we investigate how to integrate more powerful LDM priors with EM frameworks in blind inversion tasks and demonstrate new state-of-the-art results.

**Diffusion Models for 3D Inverse Problems.** 3D reconstruction from 2D images, also known as inverse graphics, has long been a significant goal in the fields of vision and graphics (Loper & Black, 2014; Chen et al., 2019; Mildenhall et al., 2020). Recently, diffusion models are also largely involved in tackling this problem (Poole et al., 2022; Lin et al., 2023; Müller et al., 2023; Shi et al., 2023b; Liu et al., 2023b). In this context, the underlying signals $\boldsymbol{x}$ and the observation $\boldsymbol{y}$ represent 3D data and 2D images, while $\mathcal{A}$ denotes the forward rendering process and $\phi$ are the camera parameters. Although the most straightforward way is to directly model 3D distributions (Müller et al., 2023; Zeng et al., 2022), this way is not feasible due to the scarcity of 3D data (Chang et al., 2015; Deitke et al., 2023). Alternatively, recent works focus on utilizing 2D diffusion priors to recover 3D scenes with SDS loss (Poole et al., 2022) but suffer from view inconsistency issues (Lin et al., 2023; Tang et al., 2023; Wang et al., 2024; Chen et al., 2023).

To mitigate this problem, a branch of work fine-tunes Latent Diffusion Models (LDMs) with multi-view images, transforming LDMs into conditional renderers (Liu et al., 2023b; Shi et al., 2023b; Tewari et al., 2023). Given an image and its camera parameter, they predict the corresponding novel views of the same 3D object. In other words, these models can also be utilized to provide 3D data priors. However, existing methods typically operate in a feed-forward fashion, still leading to accumulated inconsistency during novel view synthesis and requiring further correction designs (Shi et al., 2023a; Liu et al., 2024; 2023a). In contrast, LatentDEM treats the sparse-view 3D reconstruction (Jiang et al., 2023) task as a blind inverse problem. Given sparse-view input images without knowing their poses, we apply Zero123 (Liu et al., 2023b) priors to jointly optimize their relative camera parameters and synthesize new views. Our method utilizes information of all input views (Song et al., 2023) and produces significantly better view-consistent objects compared to feed-forward baselines (Liu et al., 2023b; Jiang et al., 2023).

## 3 PRELIMINARY

**Diffusion Models and Latent Diffusion Models.** DMs (Ho et al., 2020; Song et al., 2021b; Sohl-Dickstein et al., 2015) model data distribution by learning the time-dependent score function $\nabla_{\boldsymbol{x}_t} \log p_t(\boldsymbol{x}_t)$ with a parameterized neural networks $\boldsymbol{s}_\theta$. In the forward step, it progressively injects noise into data through a forward-time SDE; while in the inverse step, it generates data from noise through a reverse-time SDE (Song et al., 2021b):

$$\text{Forward-time SDE:} \quad \mathrm{d}\boldsymbol{x} = -\frac{\beta_t}{2}\boldsymbol{x}\mathrm{d}t + \sqrt{\beta_t}\mathrm{d}\boldsymbol{w},$$

$$\text{Reverse-time SDE:} \quad \mathrm{d}\boldsymbol{x} = \left[-\frac{\beta_t}{2}\boldsymbol{x} - \beta_t \nabla_{\boldsymbol{x}_t} \log p_t(\boldsymbol{x}_t)\right]\mathrm{d}t + \sqrt{\beta_t}\mathrm{d}\overline{\boldsymbol{w}}, \quad (2)$$

where $\beta_t \in (0,1)$ is the noise schedule, $t \in [0, T]$, $\boldsymbol{w}$ and $\overline{\boldsymbol{w}}$ are the standard Wiener process running forward and backward in time, respectively. This equation is also called variance-preserving SDE (VP-

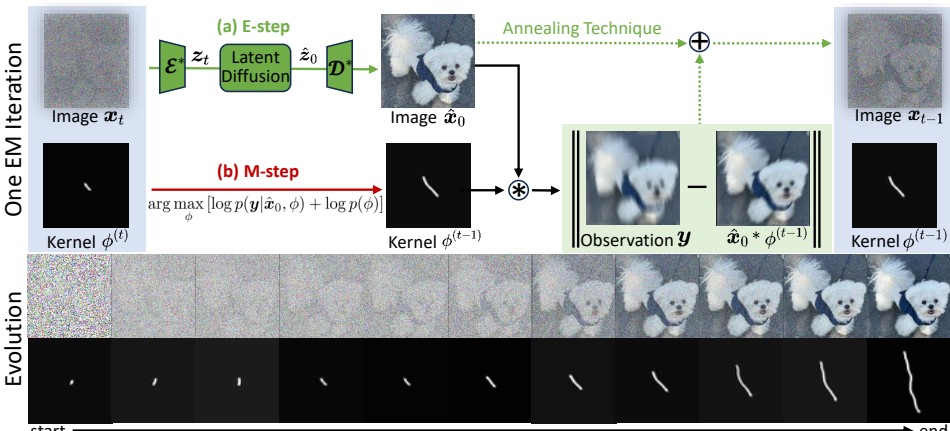

Figure 2: **Overview of LatentDEM. Top**: One EM iteration. Given currently estimated data and kernel, in the E-step, we draw new samples with LDM priors with the proposed *annealing technique*. In the M-step, we apply the maximum-a-posterior (MAP) algorithm to update forward parameters. **Bottom**: Evolution of the optimized signals and forward parameters.

SDE) that equals DDPM (Ho et al., 2020). Through this paper, we define $\alpha_t := 1 - \beta_t, \bar{\alpha}_t := \prod_{i=1}^t \alpha_i$ following Ho et al. (2020), as adopted in Algorithm 1, 2.

A significant drawback of pixel-based DMs is that they require substantial computational resources and a large volume of training data. To reduce the computation overhead, a generalized family of Latent Diffusion Models (LDMs) is proposed (Rombach et al., 2022; Blattmann et al., 2023). LDMs embed data into a compressed latent space through $z = \mathcal{E}(x)$, model the diffusion process of $z$ for efficiency and flexibility, and decode the latent code $z$ back to the pixel space through $x = \mathcal{D}(z)$, where $\mathcal{E} : \mathbb{R}^D \to \mathbb{R}^N$ and $\mathcal{D} : \mathbb{R}^N \to \mathbb{R}^D$ are the encoder and decoder, respectively. LDMs fuel state-of-the-art foundation models such as Stable Diffusion (Rombach et al., 2022), which can serve as a powerful cross-domain prior. The versatility of LDMs makes them promising solvers for inverse problems. However, such an efficient paradigm is a double-edged sword, as LDMs are notorious for their instability due to the vulnerability of latent space (Rout et al., 2024; Chung et al., 2023).

**Diffusion Models for Inverse Problems.** A common approach to apply DM priors in non-blind inverse problems is to replace unconditional score function $\nabla_{x_t} \log p_t(x_t)$ with conditional score function $\nabla_{x_t} \log p_t(x_t|y)$ and apply posterior sampling (Song et al., 2021a; Chung et al., 2022b). With the Bayesian rules, we have

$$\nabla_{x_t} \log p_t(x_t|y) = \nabla_{x_t} \log p_t(x_t) + \nabla_{x_t} \log p_t(y|x_t) \tag{3}$$

$$\approx s_\theta(x_t, t) + \log p(y|\hat{x}_0(x_t)). \tag{4}$$

While $\nabla_{x_t} \log p_t(x_t)$(Eq. 3 middle) can be approximated by diffusion models $s_\theta(x_t, t)$(Eq. 4 left). However, $\nabla_{x_t} \log p_t(y|x_t)$(Eq. 3 right) is not tractable as the likelihood $p_t(y|x_t)$ is not known when $t \neq 0$. Following DPS (Chung et al., 2022b), we also assume $p_t(y|x_t) = \int_{x_0} p(y|x_0)p(x_0|x_t)\mathrm{d}x_0 \approx p(y|\hat{x}_0(x_t))$, where $\hat{x}_0(x_t) = \mathbb{E}[x_0|x_t]$, which is computational efficient and yields reasonable results. We apply the same trick (Rout et al., 2024) for the latent space sampling as well.

**Expectation-Maximum Algorithm.** The Expectation-Maximization (EM) algorithm (Dempster et al., 1977; Gao et al., 2021) is an iterative optimization method used to estimate the parameters $\phi$ of the statistical models that involve underlying variables $x$ given the observations $y$. It aims to maximize the data likelihood $\log p_\phi(y)$. Through Jensen's inequality, this MLE can be simplified as maximizing its lower bound, equivalent to minimizing the Kullback-Leibler (KL) divergence with respect to the model parameters $\phi$ and an auxiliary distribution $q(x|y)$ (Murphy, 2023):

$$\log p_\phi(y) = \log \int q(x|y) \frac{p_\phi(y, x)}{q(x|y)} dx \geq \int q(x|y) \log \frac{p_\phi(y, x)}{q(x|y)} dx = -D_{KL}(q(x|y) \parallel p_\phi(y, x))$$

$$= -\int q(x|y) \log \frac{q(x|y)}{p_\phi(y|x)p(x)} dx = -\mathbb{E}_{q(x|y)} [\log q(x|y) - \log p_\phi(y|x) - \log p(x)] = L(q, \phi),$$

$$\tag{5}$$

where $p_\phi(\boldsymbol{x}, \boldsymbol{y})$ denotes the true joint distribution. To solve this optimization problem, the EM algorithm iterates between optimizing $q(\boldsymbol{x}|\boldsymbol{y})$ and $\phi$, known as the Expectation step (E-step) and the Maximization step (M-step), respectively. In the E-step, $q(\boldsymbol{x}|\boldsymbol{y})$ is optimized if and only if the equality holds in Eq. 5, which, according to the Jensen's inequality, require $\boldsymbol{x}$ to be sampled from $p_\phi(\boldsymbol{x}|\boldsymbol{y})$, assuming the model parameters $\phi$ are known. In the M-step, $\phi$ is optimized using the posterior samples obtained from the E-step. Through this iterative approach, the EM algorithm converges towards a local maximum of the observed data log-likelihood, making it a versatile tool for estimation problems with underlying variables. The step-by-step derivation is provided in Appendix G.

## 4 METHOD

We now describe how to solve *blind* inverse problems using latent diffusion priors. Our method is formulated within a variational Expectation-Maximization (EM) framework, where the signals and forward operators are iteratively optimized through EM iterations. In the E-step (Sec.4.1), we leverage latent diffusion priors to draw signal posterior samples, where we introduce an annealing technique to stabilize the optimization process. In the M-step (Sec.4.2), we estimate forward operators in a maximum-a-posteriori (MAP) manner, and adopt a skip-gradient method to improve the efficiency. In Sec.4.3, we show how our framework can be applied to solve representative problems such as 2D blind deblurring and 3D pose-free sparse-view reconstruction.

### 4.1 E-STEP: POSTERIOR SAMPLING VIA LATENT DIFFUSION

The goal of LatentDEM's E-step is to solve for the posterior distribution $p_\phi(\boldsymbol{x}|\boldsymbol{y})$ by leveraging the reverse time SDE in Eq. 2. To utilize the latent priors, inspired by PSLD (Rout et al., 2024), we conduct posterior sampling in the latent space by defining a conditional latent diffusion process:

$$\nabla_{\boldsymbol{z}_t} \log p_t(\boldsymbol{z}_t|\boldsymbol{y}) = \nabla_{\boldsymbol{z}_t} \log p_t(\boldsymbol{z}_t) + \nabla_{\boldsymbol{z}_t} \log p_t(\boldsymbol{y}|\boldsymbol{z}_t) \tag{6}$$

$$\approx \boldsymbol{s}_\theta^*(\boldsymbol{z}_t, t) + \nabla_{\boldsymbol{z}_t} \log p_\phi(\boldsymbol{y}|\mathcal{D}^*(\mathbb{E}[\boldsymbol{z}_0|\boldsymbol{z}_t])) \tag{7}$$

$$= \boldsymbol{s}_\theta^*(\boldsymbol{z}_t, t) - \frac{1}{2\sigma^2} \nabla_{\boldsymbol{z}_t} \|\boldsymbol{y} - \mathcal{A}_\phi(\mathcal{D}^*(\mathbb{E}[\boldsymbol{z}_0|\boldsymbol{z}_t]))\|_2^2 \tag{8}$$

where $\boldsymbol{s}_\theta^*(\boldsymbol{z}_t, t)$ is the pre-trained LDM that approximates the latent score function $\nabla_{\boldsymbol{z}_t} \log p_t(\boldsymbol{z}_t)$, $\mathcal{A}_\phi$ is the parameterized forward model, $\sigma$ is the standard deviation of the additive observation noise, $\mathcal{D}^*$ is the pre-trained latent decoder, and $\mathbb{E}[\boldsymbol{z}_0|\boldsymbol{z}_t]$ can be estimated through a reverse time SDE from $\boldsymbol{z}_t$ (Chung et al., 2022b). However, Eq. (8) works only when $\mathcal{A}_\phi$ is known, *i.e.*, non-blind settings (Rout et al., 2024). In the context of blind inversion, $\mathcal{A}_\phi$ is randomly initialized so that significant modeling errors perturb the optimization of $\boldsymbol{z}_t$ in the latent space. Consequently, there are significant artifacts when directly applying Eq. 8. We have to introduce an annealing technique to stabilize the training process:

**Technique 1 (Annealing consistency)** *Suppose that the estimated forward operator $\mathcal{A}_\phi$ is optimized from coarse to fine in the iterations, we have that:*

$$\nabla_{\boldsymbol{z}_t} \log p_t(\boldsymbol{z}_t|\boldsymbol{y}) \approx \boldsymbol{s}_\theta^*(\boldsymbol{z}_t, t) - \frac{1}{2\zeta_t\sigma^2} \nabla_{\boldsymbol{z}_t} \|\boldsymbol{y} - \mathcal{A}_\phi(\mathcal{D}^*(\mathbb{E}[\boldsymbol{z}_0|\boldsymbol{z}_t]))\|_2^2, \tag{9}$$

*where $\zeta_t$ is a time-dependent factor that decreases over time,* e.g.*, it anneals linearly from 10 at $t = 1000$ to 1 at $t = 600$ and then holds.*

We refer to this scaling technique as *Annealing consistency*. Intuitively, $\mathcal{A}_\phi$ is randomly initialized at the beginning, which cannot provide correct gradient directions. Therefore, we reduce its influence on the evolution of $\boldsymbol{z}_t$ with a large factor ($\zeta_t = 10$). As sampling progresses, $\mathcal{A}_\phi$ gradually aligns with the underlying true forward operator. We then anneal the factor ($\zeta_t = 1$) to enforce data consistency. We find that this annealing technique is critical for blind inversion with latent priors; without it, the optimized signal $\boldsymbol{x}$ consistently exhibits severe artifacts, as shown in Figure 5. Further theoretical explanations can be found in Appendix B.

### 4.2 M-STEP: FORWARD OPERATOR ESTIMATION

The goal of LatentDEM's M-step is to update the forward operator parameters $\phi$ with the estimated samples $\hat{\boldsymbol{x}}_0$ from the E-step. This can be achieved by solving a maximum-a-posterior (MAP)

---

**Algorithm 1** LatentDEM for Blind Deblurring

---

**Require:** $T, \boldsymbol{y}, \{\zeta_i\}_{i=1}^T, \{\bar{\alpha}_t\}_{t=1}^T, \{\tilde{\sigma}_t\}_{t=1}^T, \sigma, \delta, \lambda, \mathcal{E}^*, \mathcal{D}^*, \boldsymbol{s}_\theta^*, K, S_T$

    $\boldsymbol{z}_T \sim \mathcal{N}(\boldsymbol{0}, \boldsymbol{I})$
    **for** $t = T$ **to** $0$ **do**
       $\boldsymbol{s} \leftarrow \boldsymbol{s}_\theta^*(\boldsymbol{z}_t, t)$
       $\hat{\boldsymbol{z}}_0 \leftarrow \frac{1}{\sqrt{\bar{\alpha}_t}} \left( \boldsymbol{z}_t + \sqrt{1 - \bar{\alpha}_t} \boldsymbol{s} \right),$
       $\boldsymbol{\epsilon} \sim \mathcal{N}(\boldsymbol{0}, \boldsymbol{I})$
       $\boldsymbol{z}_{t-1} \leftarrow \sqrt{\bar{\alpha}_{t-1}} \hat{\boldsymbol{z}}_0 + \sqrt{1 - \bar{\alpha}_{t-1} - \tilde{\sigma}_t^2} \boldsymbol{s} + \tilde{\sigma}_t \boldsymbol{\epsilon}$
       **if** $(t > S_T$ **and** $t \mid K)$ **or** $t < S_T$ **then**
          $\hat{\boldsymbol{x}}_0 = \mathcal{D}^*(\hat{\boldsymbol{z}}_0)$                                     ▷ Skip gradient
          $\mathcal{A}_{\phi^{(t-1)}} = \text{M-step}(\boldsymbol{y}, \hat{\boldsymbol{x}}_0, \mathcal{A}_{\phi^{(t)}}, \lambda, \delta)$
          $\boldsymbol{z}_{t-1} \leftarrow \boldsymbol{z}_{t-1} - \frac{1}{2\zeta_t \sigma^2} \nabla_{\boldsymbol{z}_t} \| \boldsymbol{y} - \mathcal{A}_{\phi^{(t-1)}}(\hat{\boldsymbol{x}}_0) \|_2^2$     ▷ Annealing consistency
       **end if**
    **end for**
    **return** $\hat{\boldsymbol{x}}_0, \hat{\mathcal{A}}_\phi$

---

estimation problem:

$$\phi^* = \arg\max_\phi \mathbb{E}_{\hat{\boldsymbol{x}}_0} \left[ \log p_\phi(\boldsymbol{y}|\hat{\boldsymbol{x}}_0) + \log p(\phi) \right] = \arg\min_\phi \mathbb{E}_{\hat{\boldsymbol{x}}_0} \left[ ||\boldsymbol{y} - \mathcal{A}_\phi(\hat{\boldsymbol{x}}_0)||_2^2 + \mathcal{R}(\phi) \right], \quad (10)$$

where $p(\phi)$ is the prior distribution of $\phi$, $\hat{\boldsymbol{x}}_0 = \mathcal{D}^*(\hat{\boldsymbol{z}}_0) = \mathcal{D}^*(\mathbb{E}[\boldsymbol{z}_0|\boldsymbol{z}_t])$. $\mathcal{R}(\phi)$ is a regularizer equivalent to $\log p(\phi)$, including sparsity, patch-based priors, plug-and-play denoisers (Pan et al., 2016; Sun et al., 2013), etc. This MAP estimation problem can be solved using either gradient-based optimization (Laroche et al., 2024) or neural networks (Chung et al., 2022a). Compared to BlindDPS (Chung et al., 2022a), which jointly optimizes $\boldsymbol{x}$ and $\phi$ using two diffusion processes, our method leverages the properties of EM (Gao et al., 2021), resulting in faster convergence and better performance.

Different from pixel-space diffusion models, latent diffusion models require encoding and decoding operations that map between latent space and pixel space (*i.e.*, $\boldsymbol{x} = \mathcal{D}^*(\boldsymbol{z})$ and $\boldsymbol{z} = \mathcal{E}^*(\boldsymbol{x})$), which takes primary time consumption. Therefore, we further design an acceleration method that "skips" these operations to improve the efficiency of LatentDEM. Specifically, in the whole EM iteration, the E-step comprises two sub-steps: prior-based diffusion ($\nabla_{\boldsymbol{z}_t} \log p_t(\boldsymbol{z}_t)$) and data likelihood-based diffusion ($\nabla_{\boldsymbol{z}_t} \log p_t(\boldsymbol{y}|\boldsymbol{z}_t)$). (See Eq. 6 for the two terms). The former happens in latent space($\nabla_{\boldsymbol{z}_t} \log p_t(\boldsymbol{z}_t)$) while the latter happens in pixel space that requires encoder-decoder operations. Moreover, the M-step also involves the encoder-decoder operations. We propose to skip these operations to accelerate the training process:

**Technique 2 (Skip gradient)** *In early stages ($t > S_T$), performing $K$ times E-step in latent space with $\nabla_{\boldsymbol{z}_t} \log p_t(\boldsymbol{z}_t)$ only, then perform the whole EM-step in both latent space and image space.*

We refer to this new technique as *Skip gradient*. We find it largely accelerates the training process without hurting the performance for two reasons. First, similar to the annealing case, in the early stages of the diffusion posterior sampling process, the sampled data $\hat{\boldsymbol{x}}_0$ and forward parameters $\phi$ are far from the true value, making frequent LDM encoding and decoding unnecessary, as they won't provide useful gradient signals. Second, while the skip-gradient steps rely only on unconditional latent diffusion ($\nabla_{\boldsymbol{z}_t} \log p_t(\boldsymbol{z}_t)$), the optimization still partially follows the previous conditional sampling trajectory, leading to meaningful convergence, as also noted in (Song et al., 2023).

We typically set $S_T = 500$ and $K = 8$, which means the total skipped number $M = (T - S_T)(1 - 1/K) = (1000 - 500)(1 - 1/8) \approx 437$ full gradient computation steps. We show it significantly reduces computation overhead while keeping PSNR values approximate to the non-skip version, as demonstrated in Table 2. Our full algorithm is described in Algorithm 1.

### 4.3 BLINDING INVERSION TASKS

Our framework incorporates powerful LDM priors within the EM framework, which enables solving both linear and non-linear inverse problems. We showcase two representative tasks: the 2D blind debluring task, and the high-dimension, non-linear 3D blind inverse rendering problem.

**2D Blind Deblurring.** In the blind deblurring task, we aim to jointly estimate the clean image $\boldsymbol{x}$ and the blurring kernel $\phi$ given a blurry observation $\boldsymbol{y} = \mathcal{A}_\phi(\boldsymbol{x}) = \phi * \boldsymbol{x}$. The LatentDEM approach proceeds as follows:

- E-step: Assuming a known blurring kernel $\phi$, sample the latent code $\boldsymbol{z}_t$ and the corresponding image $\hat{\boldsymbol{x}}_0^{(t)} = \mathcal{D}(\hat{\boldsymbol{z}}_0(\boldsymbol{z}_t))$ based on Eq. 8. To enhance training stability, we adopt the "gluing" regularization (Rout et al., 2024) to address the non-injective nature of the latent-to-pixel space mapping. More discussions about this regularization are shown in Appendix C.

- M-step: Estimate blur kernels using Half-Quadratic Splitting (HQS) optimization (Geman & Yang, 1995; Laroche et al., 2024):

$$\mathbf{Z}^{(t-1)} = \mathcal{F}^{-1}\left(\frac{\mathcal{F}(\boldsymbol{y})\sum_{i=1}^n \overline{\mathcal{F}(\hat{\boldsymbol{x}}_0^{(t)})} + n\delta\sigma^2\mathcal{F}(\phi^{(t)})}{\sum_{i=1}^n \mathcal{F}(\hat{\boldsymbol{x}}_0^{(t)})\overline{\mathcal{F}(\hat{\boldsymbol{x}}_0^{(t)})} + n\delta\sigma^2}\right), \quad \phi^{(t-1)} = \mathbf{D}_{\sqrt{\lambda/\delta}}(\mathbf{Z}^{(t-1)}), \quad (11)$$

where $\mathbf{Z}$ is an intermediate variable, $\mathbf{D}$ is a Plug-and-Play (PnP) neural denoiser (Laroche et al., 2024; Zhang et al., 2017), $\mathcal{F}$ and $\mathcal{F}^{-1}$ are forward and inverse Fourier transforms, $\sigma$ defines the noise level of measurements, and $\lambda, \delta$ are tunable hyperparameters (Zhang et al., 2021). The superscripts $^{(t-1)}$ and $^{(t)}$ index diffusion steps, and $n$ is the number of samples. More details on implementation are provided in Appendix C.

**Pose-free Sparse-view 3D Reconstruction.** We also demonstrate for the first time that LDM-based blind inversion can be applied to sparse-view, unposed 3D reconstruction, a challenging task that jointly reconstructs the 3D object and camera parameters. Zero123, a conditional LDM, is utilized to approximate the 3D diffusion prior in our task. Given an input image $\boldsymbol{y}$ and camera parameters $\phi = (R, T)$ at a target view, Zero123 generates a novel-view image $\hat{\boldsymbol{x}}_0^{(t)} = \mathcal{D}(\hat{\boldsymbol{z}}_0), \hat{\boldsymbol{z}}_0 = \mathbb{E}[\boldsymbol{z}_0|\boldsymbol{z}_t]$ through a conditional latent diffusion process $\nabla_{\boldsymbol{z}_t} \log p_t(\boldsymbol{z}_t|\boldsymbol{y}, \phi)$. However, the current Zero123 is limited to view synthesis and 3D generation from a single image.

By integrating Zero123 into LatentDEM, we can reconstruct a view-consistent 3D object from multiple unposed images. Without loss of generality, we illustrate this with two images $\boldsymbol{y}_1$ and $\boldsymbol{y}_2$ without knowing their relative pose. The LatentDEM approach becomes:

- E-step: Assuming known camera parameters $\phi_1$ and $\phi_2$, aggregate information through a joint latent diffusion process $\nabla_{\boldsymbol{z}_t} \log p_t(\boldsymbol{z}_t|\boldsymbol{y}_1, \phi_1, \boldsymbol{y}_2, \phi_2)$ to create view-consistent latent codes $\boldsymbol{z}_t$ and synthesized image $\hat{\boldsymbol{x}}_0^{(t)}$.

- M-step: Estimate camera parameters based on $\hat{\boldsymbol{x}}_0^{(t)}$ by aligning unposed images to synthetic and reference views via gradient-based optimization:

$$\phi_2^{(t-1)} = \phi_2^{(t)} - \lambda\nabla_{\phi_2^{(t)}}\|\boldsymbol{z}_t(\boldsymbol{y}_2, \phi_2^{(t)}) - \boldsymbol{z}_t(\hat{\boldsymbol{x}}_0^{(t)}, \mathbf{0})\|_2^2 - \delta\nabla_{\phi_2^{(t)}}\|\boldsymbol{z}_t(\boldsymbol{y}_2, \phi_2^{(t)}) - \boldsymbol{z}_t(\boldsymbol{y}_1, \phi_1)\|_2^2, \quad (12)$$

where $\boldsymbol{z}_t(\cdot, \cdot)$ represents the time-dependent latent features of an image after specified camera transformation, $\mathbf{0}$ indicates no transformation, and $\lambda, \delta$ are tunable hyperparameters. Note only $\phi_2$ is optimized, as $\phi_1$ defines the reference view.

Through the synthesis of multiple random novel views from input images and subsequent volumetric rendering, we finally generate a comprehensive 3D representation of the object. This approach extends to arbitrary n unposed images, where n-1 camera poses should be estimated. More views yield better 3D generation/reconstruction performance. It outperforms state-of-the-art pose-free sparse-view 3D baselines (Jiang et al., 2023) and generates well-aligned images for detailed 3D modeling (Liu et al., 2024). Further details, including the derivation of the view-consistent diffusion process from traditional Zero123 and 3D reconstruction results from various numbers of images, are provided in Appendix D.

## 5 EXPERIMENTS

In this section, we first apply our method on the 2D blind deblurring task in Sec. 5.1. We then demonstrate our method on pose-free, sparse-view 3D reconstruction in Sec. 5.2. Lastly, we perform extensive ablation studies to demonstrate the proposed techniques. Additional implementation details and results can be found in Appendix C, D and E.

Figure 3: **Blind motion deblurring results**. Row (1-2): ImageNet. Row (3-4): FFHQ. Our method successfully recovers clean images and accurate blur kernels, consistently outperforming all the baselines, even under challenging cases where the observations are severely degraded.

## 5.1 2D BLIND MOTION DEBLURRING

**Dataset.** We evaluate our method on the images from the widely used ImageNet (Deng et al., 2009) and FFHQ (Karras et al., 2019). We randomly choose 64 validation images from each dataset, where the resolutions are both $256 \times 256$. We chose to use the state-of-the-art Stable Diffusion v-1.5 model (Rombach et al., 2022) as our cross-domain prior. For quantitative comparison, we evaluate the image quality with three metrics: peak signal-to-noise-ratio (PSNR), structural similarity index (SSIM), and learned perceptual image patch similarity (LPIPS). We assess the estimated kernel via mean-squared error (MSE), and maximum of normalized convolution (MNC) (Hu & Yang, 2012). We also provide a comparison with SOTA self-supervised methods on an additional standard benchmark (Lai et al., 2016), which is shown in Appendix E.

Table 1: Quantitative evaluation (PSNR, SSIM, LPIPS) of blind deblurring task and (MSE, MNC) of kernel estimation on ImageNet and FFHQ. **Bold**: Best, under: second best.

| Method | ImageNet ($256 \times 256$) | | | | | FFHQ ($256 \times 256$) | | | | |
|---|---|---|---|---|---|---|---|---|---|---|
| | Image | | | Kernel | | Image | | | Kernel | |
| | PSNR ↑ | SSIM ↑ | LPIPS ↓ | MSE ↓ | MNC ↑ | PSNR ↑ | SSIM ↑ | LPIPS ↓ | MSE ↓ | MNC ↑ |
| MPRNet | **19.85** | 0.433 | 0.470 | - | - | 21.60 | 0.517 | 0.399 | - | - |
| Self-Deblur | 16.74 | 0.232 | 0.493 | 0.016 | 0.036 | 18.84 | 0.328 | 0.493 | 0.017 | 0.045 |
| BlindDPS | 17.31 | 0.472 | 0.309 | 0.036 | 0.274 | 22.58 | 0.583 | 0.245 | 0.048 | 0.270 |
| FastEM | 17.36 | 0.422 | 0.377 | 0.440 | 0.266 | 17.46 | 0.554 | 0.169 | 0.035 | 0.399 |
| **Ours** | 19.35 | **0.496** | **0.256** | **0.010** | **0.441** | **22.65** | **0.653** | **0.167** | **0.009** | **0.459** |

**Results.** We provide motion deblurring results in Figure 3 and Table 1. Our method is compared with two state-of-the-art methods that directly apply pixel-space diffusion models for blind deblurring: BlindDPS (Chung et al., 2022a) and FastEM (Laroche et al., 2024), and three widely applied methods: MPRNet (Zamir et al., 2021), DeblurGAN V2 (Kupyn et al., 2019), and Self-Deblur (Ren et al., 2020). Several interesting observations can be found here. First, LatentDEM outperforms all the baselines qualitatively. As shown in Fig. 3, in challenging cases with severe motion blur and aggressive image degradation, previous methods are unable to accurately estimate the kernel, while the proposed method enables accurate kernel estimation and high-quality image restoration. We attribute this to the fact that the powerful LDM priors provide better guidance than pixel-space DM priors in the posterior sampling, together with the deliberately designed EM optimization policies. Moreover, as

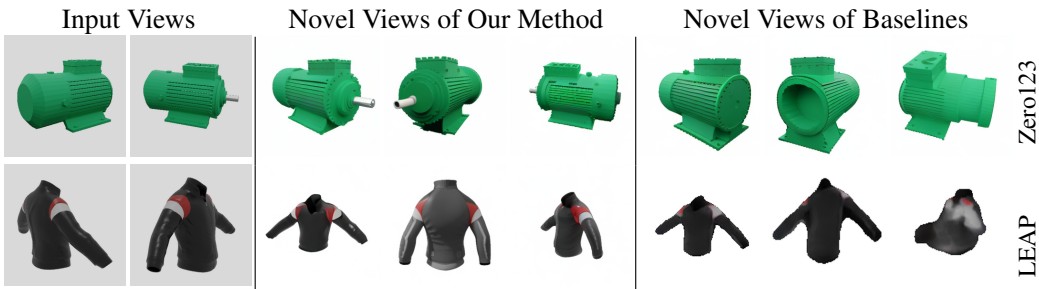

Figure 4: **Pose-free sparse-view 3D reconstruction results.** Our method successfully synthesizes consistent novel view images given two sparse input views. In contrast, Zero123 (Liu et al., 2023b) produces images missing the engine handle that are not consistent with the input views, while LEAP (Jiang et al., 2023) fails to generate photo-realistic images.

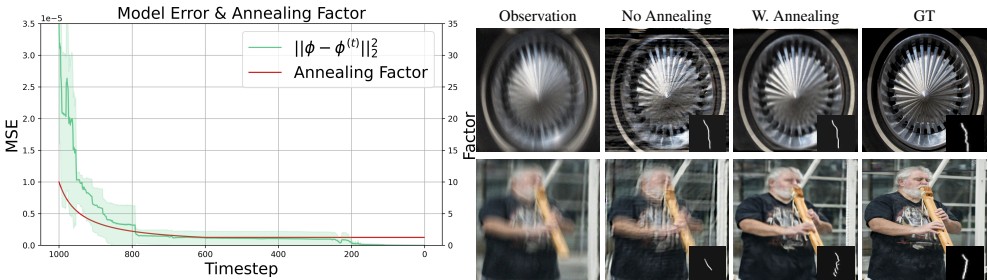

Figure 5: **Effectiveness of our annealed consistency technique.** Left: blur kernel accuracy curve (green) on 10 examples (std are represented by shadow). It indicates that the kernel is very wrong at the beginning but becomes meaningful when $t < 600$, which corresponds to the annealing factor curve (red). Right: we further show that simply applying LDM priors in blind inversion produces images with severe artifacts due to the fragile latent space, while the annealing technique stabilizes the optimization and generates much better results. Gluing term is used in all experiments.

shown in Table 1, LatentDEM also achieves the best scores in most metric evaluations, especially for kernel estimation, which demonstrates the efficacy of our EM framework. Interestingly, MPRNet shows higher PSNR in ImageNet dataset but visually it produces smooth and blurry results, which indicates the quality of deblurring cannot be well-reflected by PSNR. Nevertheless, LatentDEM still largely outperforms in SSIM and LPIPS metrics. We additionally compared LatentDEM with the current SOTA self-supervised deblurring method, using datasets from the self-supervised deblurring benchmark ((Lai et al., 2016)). The results are in Table 4, 5 and Fig. 10. In most cases, LatentDEM outperforms (Li et al., 2023) without the need for heavy parameter tuning.

### 5.2 EXPERIMENTS ON POSE-FREE SPARSE-VIEW 3D RECONSTRUCTION

**Dataset.** We evaluate the pose-free sparse-view 3D reconstruction performance on Objaverse dataset (Deitke et al., 2023), which contains millions of high-quality 3D objects. We pick up 20 models and for each model, we randomly render two views without knowing their poses. Our goal is to synthesize novel views and reconstruct the underlying 3D model from the unposed sparse-view inputs, which is a very challenging task (Jiang et al., 2023) and cannot be easily addressed by NeRF (Mildenhall et al., 2020) or Gaussian Splatting (Kerbl et al., 2023) that require image poses.

**Results.** We provide novel view synthesis results in Fig 4. Our model is built on top of Zero123 (Liu et al., 2023b) priors. Zero123 has demonstrated effectiveness in synthesizing high-quality novel-view images, but it sometimes fails in creating view-consistent results across different views, as shown in the first row of Fig. 4. A major reason is that it synthesizes new views from a single input only and cannot capture the whole information of the 3D object, resulting in hallucinated synthesis. Instead, our method could easily crack this nut by aggregating the information of all input views and embedding them together through hard data consistency (Song et al., 2023). We show that multiple consistent novel view images can be acquired from only two input views. Moreover, our method doesn't need to know the poses of images as they can be jointly estimated, which supports the more

Table 2: **Effectiveness of our skip gradient technique.** We evaluate different skip gradient schemes on the blind deblurring task. Compared to the setting without skipping, skipping half steps or even more steps performs similarly while reducing the running time significantly.

| | Method | Running Time | Image PNSR | Kernel MSE |
|---|---|---|---|---|
| a | FastEM | 1min30sec | 17.46 | 0.048 |
| b | BlindDPS | 1min34sec | 22.58 | 0.035 |
| c | No skipping. ($S_T = 1000, K = 1, M = 0$) | 6min33sec | 23.45 | 0.011 |
| d | Skip-grad. ($S_T = 500, K = 8, M = 437$) | 4min17sec | 23.44 | 0.011 |
| e | Skip-grad. ($S_T = 0, K = 8, M = 875$) | 2min54sec | 23.00 | 0.010 |
| f | Skip-grad. ($S_T = 0, K = 16, M = 937$) | 1min20sec | 22.56 | 0.010 |

challenging pose-free, sparse view 3D reconstruction. We compare our method with the current state-of-the-art pose-free 3D reconstruction baseline, LEAP (Jiang et al., 2023). As shown in the second row of Fig. 4, while LEAP fails to generate photo-realistic new views, our method could leverage the powerful Zero123 prior to overcome texture degradation and geometric misalignment, maintaining fine details like the geometry and the texture of the jacket. Besides, we find that adding more views significantly improves the fidelity of the 3D reconstruction. The LatentDEM framework facilitates consistent 3D reconstruction across different images. We provide more results and analysis in Appendix D.4, D.5, E.

### 5.3 ABLATION STUDIES

**Annealed Consistency.** A major problem when using LDMs instead of pixel-based DMs is the vulnerable latent space optimization. In the context of blind inverse problems, the inaccurate forward operators at the beginning could make the problem even worse, where the optimal solutions significantly deviate from the true value and contain strong image artifacts, as demonstrated in Fig. 5 right. To ensure stable optimization, we should set our empirical annealing coefficients($\zeta_t$ anneals linearly from 10 at $t = 1000$ to 1 at $t = 600$ and then holds) based on the forward modeling errors, as shown in Fig. 5. This technique show stabilizes the optimization process and produces a more accurate estimation (Fig. 5 right). We provide more annealing analysis in Appendix B, which explains why annealing consistency aligns better with the blind inversion problem and brings more stable optimization, as well as superior performance.

**Skip Gradient for Acceleration.** We also investigate the influence of the skip-gradient technique, where we compute the full EM step every $K$ steps while in the middle steps we only run the latent space diffusion. We validate 4 different groups of hyperparameters and compare their running times and imaging quality on 2D blind deblurring task (Table 2). We find the running time for LatentDEM linearly decreases as the number of accelerated steps $M$ increases. In the accelerated steps, the encoder-decoder inference and gradients are skipped, therefore significantly reducing the total optimization time. Moreover, though we have skipped a lot of computation burden, due to the fact that the diffusion tends to follow the previous optimization trajectory, it still results in meaningful convergence. In an extreme setting (case f) where we skip 900 gradients, our method still outperforms baselines, as well as achieving the fastest optimization speed.

## 6 CONCLUSION

In this work, we proposed LatentDEM, a novel method that incorporates powerful latent diffusion priors for solving blind inverse problems. Our method jointly estimates underlying signals and forward operator parameters using an EM framework with two carefully designed optimization techniques, which reveals more accurate and efficient 2D blind debluring performance than prior arts, as well as demonstrates new capabilities in 3D pose-free sparse-view reconstruction. Its limitation includes that in 3D tasks it still relies on LDMs fine-tuned with multi-view images. It is interesting to think about how to combine LatentDEM with SDS loss to directly run 3D inference from purely 2D diffusion models. The code and project page are available upon request.

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

## A  APPENDIX

We provide an appendix to describe the details of our derivations and algorithms, as well as show more results. We first provide a theoretical explanation for the annealing consistency technique in Sec. B. We then provide implementation details of the 2D blind deblurring in Sec. C and the 3D pose-free, sparse-view reconstruction in Sec. D. Lastly, we report more results in Sec. E.

## B  THEORETICAL EXPLANATIONS FOR ANNEALING CONSISTENCY

We consider the blind inversion as a time-dependent modeling process and the annealing consistency technique helps address time-dependent modeling errors. Specifically, we express the image formation model as:

$$\boldsymbol{y} = \mathcal{A}_{\phi^{(t)}}(\hat{\boldsymbol{x}}_0) + w_t + \boldsymbol{n}, \quad w_t \sim \mathcal{N}(\boldsymbol{0}, \nu_t^2 \mathbf{I}), \quad \boldsymbol{n} \sim \mathcal{N}(\boldsymbol{0}, \sigma^2 \mathbf{I}), \tag{13}$$

where $\phi^{(t)}$ represents the estimated forward model parameters at step $t$, $\hat{\boldsymbol{x}}_0 = \mathcal{D}^*(\mathbb{E}(\boldsymbol{z}_0|\boldsymbol{z}_t))$, $w_t$ is the time-dependent modeling noise which is assumed to follow a Gaussian distribution with a time-dependent standard deviation of $\nu_t$, and $\boldsymbol{n}$ is the observation noise with a constant standard deviation of $\sigma$. Therefore, following (Chung et al., 2022b) we derive below data likelihood term to account for both modeling errors and observation noise during the diffusion posterior sampling:

$$\nabla_{\boldsymbol{z}_t} \log p_t(\boldsymbol{y}|\boldsymbol{z}_t) \approx -\frac{1}{2(\nu_t^2 + \sigma^2)} \nabla_{\boldsymbol{x}_t} \|\boldsymbol{y} - \mathcal{A}_\phi(\mathcal{D}^*(\mathbb{E}[\boldsymbol{z}_0|\boldsymbol{z}_t]))\|_2^2, \tag{14}$$

where $\nu_t^2$ should gradually decrease from a large value to zero as the estimated model parameters converge to the ground truth. This is consistent with the proposed technique, where in Eq. 9 $\zeta_t$ linearly anneals from a large number to a constant. As a result, annealing consistency aligns better with the blind inversion problem and brings more stable optimization, as well as superior performance.

## C  IMPLEMENTATION DETAILS OF 2D BLIND DEBLURRING

### C.1  E-STEP

**Diffusion Posterior Sampling with Gluing Regularization.**  In the 2D blind deblurring task, the E-step performs latent diffusion posterior sampling (DPS) to reconstruct the underlying image, assuming a known blur kernel. The basic latent DPS takes the form as follows:

$$\nabla_{\boldsymbol{z}_t} \log p_t(\boldsymbol{z}_t|\boldsymbol{y}) \approx \boldsymbol{s}_\theta^*(\boldsymbol{z}_t, t) + \nabla_{\boldsymbol{z}_t} p(\boldsymbol{y}|\mathcal{D}^*(\mathbb{E}[\boldsymbol{z}_0|\boldsymbol{z}_t])),$$
$$= \boldsymbol{s}_\theta^*(\boldsymbol{z}_t, t) - \frac{1}{2\zeta_t \sigma^2} \nabla_{\boldsymbol{z}_t} \|\boldsymbol{y} - \mathcal{A}_\phi(\mathcal{D}^*(\mathbb{E}[\boldsymbol{z}_0|\boldsymbol{z}_t]))\|_2^2, \tag{15}$$

which simply transform the equation from pixel-based DPS (Chung et al., 2022b) to the latent space. However, this basic form always produces severe artifacts or results in reconstructions inconsistent with the measurements (Song et al., 2023). A fundamental reason is that the decoder is an one-to-many mapping, where numerous latent codes $\boldsymbol{z}_0$ that represent underlying images can match the measurements. Computing the gradient of the density specified by Eq. 15 could potentially drive $\boldsymbol{z}_t$ towards multiple different directions. To address this, we employ an additional constraint called "gluing" (Rout et al., 2024) to properly guide the optimization in the latent space:

$$\nabla_{\boldsymbol{z}_t} \log p(\boldsymbol{y}|\boldsymbol{z}_t) = \underbrace{\nabla_{\boldsymbol{z}_t} p(\boldsymbol{y}|\mathcal{D}^*(\mathbb{E}[\boldsymbol{z}_0|\boldsymbol{z}_t]))}_{\text{DPS vanilla extension}}$$
$$+ \gamma \underbrace{\nabla_{\boldsymbol{z}_t} \|\mathbb{E}[\boldsymbol{z}_0|\boldsymbol{z}_t] - \mathcal{E}^*\left(\mathcal{A}_{\phi^{(t-1)}}^T \boldsymbol{y} + \left(\boldsymbol{I} - \mathcal{A}_{\phi^{(t-1)}}^T \mathcal{A}_{\phi^{(t-1)}}\right) \mathcal{D}^*(\mathbb{E}[\boldsymbol{z}_0|\boldsymbol{z}_t])\right)\|_2^2}_{\text{"gluing" regularization}}, \tag{16}$$

where $\gamma$ is a tunable hyperparameter. The gluing objective (Rout et al., 2024) is critical for LatentDEM as it constrains the latent code update following each M-step, ensuring that the denoising update, data fidelity update, and the gluing update point to the same optima. Note that gluing is also involved in the skip-gradient technique, *i.e.*, we will also ignore it during the skipped steps.

## C.2 M-STEP

The M-step solves the MAP estimation of the blur kernel using the posterior samples $\hat{x}_0$ from the E-step. This process is expressed by:

$$\phi^* = \arg \min_\phi \mathbb{E}_{\hat{x}_0} \left[ \frac{1}{2\sigma^2} ||y - \mathcal{A}_\phi(\hat{x}_0)||_2^2 + \mathcal{R}(\phi) \right], \tag{17}$$

where $\sigma^2$ denotes the noise level of the measurements and $\mathcal{R}$ is the regularizer. Common choices of the regulation term can be $l_2$ or $l_1$ regularizations on top of the physical constraints on the blur kernel (non-negative values that add up to one). Despite being quite efficient when the blurry image does not have noise, they generally fail to provide high-quality results when the noise level increases (Laroche et al., 2024). Therefore, we decide to leverage a Plug-and-Play (PnP) denoiser, $\mathbf{D}_{\sigma_d}$, as the regularizer. We find that training the denoiser on a dataset of blur kernels with various noise levels ($\sigma_d$) can lead to efficient and robust kernel estimation. Now with this PnP denoiser as the regularizer, we can solve Eq. 17 with the Half-Quadratic Splitting (HQS) optimization scheme:

$$\mathbf{Z}_{i+1} = \arg \min_{\mathbf{Z}} \left[ \frac{1}{2\sigma^2} ||\mathcal{A}_{\mathbf{Z}}(\hat{x}_0) - y||_2^2 + \frac{\delta}{2} ||\mathbf{Z} - \phi_i||_2^2 \right], \tag{18}$$

$$\phi_{i+1} = \mathbf{D}_{\sqrt{\lambda/\delta}}(\mathbf{Z}_{i+1}) = \arg \min_\phi \left[ \lambda \mathcal{R}(\phi) + \frac{\delta}{2} ||\phi - \mathbf{Z}_{i+1}||_2^2 \right], \tag{19}$$

where $\mathbf{Z}$ is a intermediate variable, $\mathbf{D}_{\sigma_d}$ is a PnP neural denoiser (Laroche et al., 2024; Zhang et al., 2017), $\sigma$ defines the noise level of measurements, and $\lambda$, $\beta$ are tunable hyperparameters (Zhang et al., 2021). The subscripts $_i$ and $_{i+1}$ index iterations of Eq. 18 and Eq. 19 in one M-step. For the deconvolution problem, Eq 18 can easily be solved in the Fourier domain and Eq 19 corresponds to the regularization step. It corresponds to the MAP estimator of a Gaussian denoising problem on the variable $\mathbf{Z}_{i+1}$. The main idea behind the PnP regularization is to replace this regularization step with a pre-trained denoiser. This substitution can be done becaue of the close relationship between the MAP and the MMSE estimator of a Gaussian denoising problem. In the end, the M-step can be expressed by Eq 11.

**Plug-and-Play Denoiser.** We train a Plug-and-Play (PnP) denoiser to serve as the kernel regularizer in the M-step. For the architecture of the denoiser, we use a simple DnCNN (Zhang et al., 2017) with 5 blocks and 32 channels. In addition to the noisy kernel, we also take the noise level map as an extra channel and feed it to the network to control the denoising intensity. The settings are similar to one of the baseline methods, FastEM (Laroche et al., 2024). In the data preparation process, we generate 60k motion deblur kernels with random intensity and add random Gaussian noise to them. The noisy level map is a 2D matrix filled with the variance and is concatenated to the kernel as an additional channel as input to the network. We train the network for 5,000 epochs by denoising the corrupted kernel and use the MSE loss. All the training is performed on a NVIDIA A100 which lasts for seven hours. We also try different network architectures like FFDNet but find the DnCNN is sufficient to tackle our task and it's very easy to train.

**Hyperparameters.** For the motion deblur task, we leverage the codebase of PSLD (Rout et al., 2024), which is based on Stable Diffusion-V1.5. Besides the hyperparameters of the annealing and skip-gradient technique, we find it critical to choose the proper parameters for the gluing and M-step. Improper parameters result in strong artifacts. In our experiments, we find the default hyperparameters in (Rout et al., 2024) won't work, potentially due to the more fragile latent space. The hyperparameters in our M-step are $\lambda = 1$ and $\delta = 5e6$, and We iterate Eq. 11 20 times to balance solution convergence and computational efficiency.

# D IMPLEMENTATION DETAILS OF POSE-FREE SPARSE-VIEW 3D RECONSTRUCTION

## D.1 BASICS

**Problem Formulation.** The pose-free, sparse-view 3D reconstruction problem aims to reconstruct a 3D object from multiple unposed images. This task can be formulated as a blind inversion problem

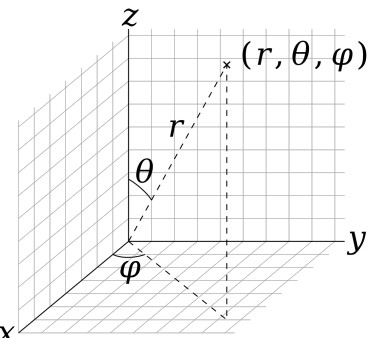

Figure 6: **Spherical coordinate system (Inc., 2008).**

with the following forward model:

$$\boldsymbol{y} = \mathcal{A}_\phi\left(\boldsymbol{x}\right), \boldsymbol{y} = \{\boldsymbol{y}_1, \cdots, \boldsymbol{y}_n\}, \quad \phi = \{\phi_1, \cdots, \phi_n\}, \tag{20}$$

where $\boldsymbol{x}$ represents the underlying 3D object, $\boldsymbol{y}$ is the set of observations containing images from multiple views, and $\phi$ denotes the camera parameters corresponding to different views. The task requires us to jointly solve for both the 3D model, $\boldsymbol{x}$, and the image poses, $\phi$, to reconstruct a view-consistent 3D object. 3D models can be represented in various forms, including meshes, point clouds, and other formats. In our paper, we implicitly represent the 3D model as a collection of random view observations of the 3D object. These views can then be converted into 3D geometry using volumetric reconstruction methods such as Neural Radiance Fields (NeRF) or 3D Gaussian Splatting (3DGS).

**Camera Model Representation.** We employ a spherical coordinate system to represent camera poses and their relative transformations. As shown in Fig. 6, we place the origin of the coordinate system at the center of the object. In this system, $\theta$, $\phi$, and $r$ represent the polar angle, azimuth angle, and radius (distance from the center to the camera position), respectively. The relative camera pose between two views is derived by directly subtracting their respective camera poses. For instance, if two images have camera poses $(\theta_1, \phi_1, r_1)$ and $(\theta_2, \phi_2, r_2)$, their relative pose is calculated as $(\theta_2 - \theta_1, \phi_2 - \phi_1, r_2 - r_1)$.

**Zero123.** Zero123 is a conditional latent diffusion model, $\nabla_{\boldsymbol{z}_t} \log p_t(\boldsymbol{z}_t|\boldsymbol{y}, \phi)$, fine-tuned from Stable Diffusion for novel view synthesis. It generates a novel-view image at a target viewpoint, $\boldsymbol{x}_0 = \mathcal{D}(\boldsymbol{z}_0)$, given an input image, $\boldsymbol{y}$, and the relative camera transformation, $\phi$, where $\mathcal{D}$ maps the latent code to pixel space. This model enables 3D object generation from a single image: by applying various camera transformations, Zero123 can synthesize multiple novel views, which can then be used to reconstruct a 3D model. As a result, Zero123 defines a powerful 3D diffusion prior.

### D.2 E-STEP

In the E-step, we perform a view-consistent diffusion process to generate target novel views from multiple input images, where we assume the camera pose of each image is known.

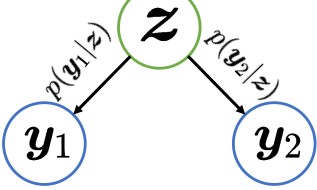

Figure 7: **Graphic model of the E-step.**

**Derivation of View-consistent Diffusion Model from Zero123.** The view-consistent generation is governed by a latent diffusion process conditioned on multiple images, defined as $\nabla_{\boldsymbol{z}_t} \log p_t(\boldsymbol{z}_t|\boldsymbol{y}_1, \phi_1, \boldsymbol{y}_2, \phi_2)$. In the graphical model of the 3D reconstruction problem, the input images $\boldsymbol{y}_1$ and $\boldsymbol{y}_2$ represent different views of the same object. These views should be independent of each other given the geometry of the 3D object, which is described by the latent code $\boldsymbol{z}_t$.

---

**Algorithm 2** LatentDEM for Pose-free Sparse-view 3D Reconstruction

---

**Require:** $T, \boldsymbol{y}_1, \boldsymbol{y}_2, \phi_1, \phi_2^{(T)}, \{\bar{\alpha}_i\}_{i=1}^T, \{\tilde{\sigma}_i\}_{i=1}^T, \{\gamma_i\}_{i=0}^{T-1}, \delta, \lambda, \mathcal{E}^*, \mathcal{D}^*, s_\theta^*$

$\quad \boldsymbol{z}_T(\boldsymbol{y}_1, \phi_1) \sim \mathcal{N}(\boldsymbol{0}, \boldsymbol{I})$

$\quad \boldsymbol{z}_T(\boldsymbol{y}_2, \phi_2^{(T)}) \sim \mathcal{N}(\boldsymbol{0}, \boldsymbol{I})$

$\quad$ **for** $t = T$ **to** $0$ **do**

$\quad\quad \boldsymbol{s}_1 \leftarrow \boldsymbol{s}_\theta^*(\boldsymbol{z}_t(\boldsymbol{y}_1, \phi_1), t, \boldsymbol{y}_1, \phi_1)$

$\quad\quad \boldsymbol{s}_2 \leftarrow \boldsymbol{s}_\theta^*(\boldsymbol{z}_t(\boldsymbol{y}_2, \phi_2^{(t)}), t, \boldsymbol{y}_2, \phi_2^{(t)})$

$\quad\quad \hat{\boldsymbol{z}}_0(\boldsymbol{y}_1, \phi_1) \leftarrow \frac{1}{\sqrt{\bar{\alpha}_i}}(\boldsymbol{z}_t(\boldsymbol{y}_1, \phi_1) + \sqrt{(1 - \bar{\alpha}_t)}\boldsymbol{s}_1)$

$\quad\quad \hat{\boldsymbol{z}}_0(\boldsymbol{y}_2, \phi_2^{(t)}) \leftarrow \frac{1}{\sqrt{\bar{\alpha}_i}}(\boldsymbol{z}_t(\boldsymbol{y}_2, \phi_2^{(t)}) + \sqrt{(1 - \bar{\alpha}_t)}\boldsymbol{s}_2)$

$\quad\quad \boldsymbol{\epsilon} \sim \mathcal{N}(\boldsymbol{0}, \boldsymbol{I})$

$\quad\quad \boldsymbol{z}_{t-1}(\boldsymbol{y}_1, \phi_1) \leftarrow \sqrt{\bar{\alpha}_{t-1}}\hat{\boldsymbol{z}}_0(\boldsymbol{y}_1, \phi_1) + \sqrt{1 - \bar{\alpha}_{t-1} - \tilde{\sigma}_t^2}\boldsymbol{s}_1 + \tilde{\sigma}_t\boldsymbol{\epsilon}$

$\quad\quad \boldsymbol{z}_{t-1}(\boldsymbol{y}_2, \phi_2) \leftarrow \sqrt{\bar{\alpha}_{t-1}}\hat{\boldsymbol{z}}_0(\boldsymbol{y}_2, \phi_2) + \sqrt{1 - \bar{\alpha}_{t-1} - \tilde{\sigma}_t^2}\boldsymbol{s}_2 + \tilde{\sigma}_t\boldsymbol{\epsilon}$

$\quad\quad \boldsymbol{z}_{t-1}(\boldsymbol{y}_1, \phi_1, \boldsymbol{y}_2, \phi_2^{(t-1)}) = \text{E-step}(\boldsymbol{z}_{t-1}(\boldsymbol{y}_1, \phi_1), \boldsymbol{z}_{t-1}(\boldsymbol{y}_2, \phi_2^{(t-1)}), \gamma_{t-1})$

$\quad\quad \phi_2^{(t-1)} = \text{M-step}(\boldsymbol{z}_{t-1}(\boldsymbol{y}_1, \phi_1), \boldsymbol{z}_{t-1}(\boldsymbol{y}_2, \phi_2^{(t)}), \boldsymbol{z}_{t-1}(\boldsymbol{y}_1, \phi_1, \boldsymbol{y}_2, \phi_2^{(t-1)}), \delta, \lambda)$

$\quad\quad \boldsymbol{z}_{t-1}(\boldsymbol{y}_1, \phi_1) \leftarrow \boldsymbol{z}_{t-1}(\boldsymbol{y}_1, \phi_1, \boldsymbol{y}_2, \phi_2^{(t-1)})$

$\quad$ **end for**

$\quad$ **return** $\mathcal{D}^*(\hat{\boldsymbol{z}}_0(\boldsymbol{y}_1, \phi_1)), \phi_2^{(0)}$

---

Consequently, the view-consistent diffusion can be derived from Zero123 as follows:

$$p(\boldsymbol{z}_{t-1}|\boldsymbol{z}_t, \boldsymbol{y}_1, \phi_1, \boldsymbol{y}_2, \phi_2) \propto p(\boldsymbol{y}_1, \boldsymbol{y}_2|\boldsymbol{z}_{t-1}, \boldsymbol{z}_t, \phi_1, \phi_2)$$
$$= p(\boldsymbol{y}_1|\boldsymbol{z}_{t-1}, \boldsymbol{z}_t, \phi_1)p(\boldsymbol{y}_2|\boldsymbol{z}_{t-1}, \boldsymbol{z}_t, \phi_2) \qquad (21)$$
$$\propto p(\boldsymbol{z}_{t-1}|\boldsymbol{z}_t, \boldsymbol{y}_1, \phi_1)p(\boldsymbol{z}_{t-1}|\boldsymbol{z}_t, \boldsymbol{y}_2, \phi_2),$$

where the conditional diffusions from single images, $p(\boldsymbol{z}_{t-1}|\boldsymbol{z}_t, \boldsymbol{y}_1, \phi_1)$ and $p(\boldsymbol{z}_{t-1}|\boldsymbol{z}_t, \boldsymbol{y}_2, \phi_2)$, are defined by Zero123, and they both follow Gaussian distributions according to the Langevin dynamics defined by the reverse-time SDE (Eq. 2), here we adopt the DDIM framework.

$$p_t(\boldsymbol{z}_{t-1}|\boldsymbol{z}_t, \boldsymbol{y}_1, \phi_1) = \mathcal{N}\left(\sqrt{\bar{\alpha}_{t-1}}\hat{\boldsymbol{z}}_0(\boldsymbol{z}_t|\boldsymbol{y}_1, \phi_1) + \sqrt{1 - \bar{\alpha}_{t-1} - \sigma_t^2} \cdot \nabla_{\boldsymbol{z}_t} \log p_t(\boldsymbol{z}_t|\boldsymbol{y}_1, \phi_1), \sigma_t^2\boldsymbol{I}\right)$$

$$p_t(\boldsymbol{z}_{t-1}|\boldsymbol{z}_t, \boldsymbol{y}_2, \phi_2) = \mathcal{N}\left(\sqrt{\bar{\alpha}_{t-1}}\hat{\boldsymbol{z}}_0(\boldsymbol{z}_t|\boldsymbol{y}_2, \phi_2) + \sqrt{1 - \bar{\alpha}_{t-1} - \sigma_t^2} \cdot \nabla_{\boldsymbol{z}_t} \log p_t(\boldsymbol{z}_t|\boldsymbol{y}_2, \phi_2), \sigma_t^2\boldsymbol{I}\right)$$
$$(22)$$

In our pose-free 3D reconstruction task, we account for potential inaccuracies in $\phi_2$ during early diffusion stages, so the diffusion process is modified as:

$$p_t(\boldsymbol{z}_{t-1}|\boldsymbol{z}_t, \boldsymbol{y}_2, \phi_2) = \mathcal{N}\left(\sqrt{\bar{\alpha}_{t-1}}\hat{\boldsymbol{z}}_0 + \sqrt{1 - \bar{\alpha}_{t-1} - \sigma_t^2} \cdot \nabla_{\boldsymbol{z}_t} \log p_t(\boldsymbol{z}_t|\boldsymbol{y}_2, \phi_2), (\sigma_t^2 + \nu_t^2)\boldsymbol{I}\right), \qquad (23)$$

where $\nu_t$ represents the standard deviation of time-dependent model errors. As a result, $p(\boldsymbol{z}_{t-1}|\boldsymbol{z}_t, \boldsymbol{y}_1, \phi_1, \boldsymbol{y}_2, \phi_2)$ becomes

$$p(\boldsymbol{z}_{t-1}|\boldsymbol{z}_t, \boldsymbol{y}_1, \phi_1, \boldsymbol{y}_2, \phi_2) = \mathcal{N}\left(\frac{\sigma_t^2 + \nu_t^2}{2\sigma_t^2 + \nu_t^2}\mu_1 + \frac{\sigma_t^2}{2\sigma_t^2 + \nu_t^2}\mu_2, \frac{\sigma_t^2(\sigma_t^2 + \nu_t^2)}{2\sigma_t^2 + \nu_t^2}\right)$$

$$\mu_1 = \sqrt{\bar{\alpha}_{t-1}}\hat{\boldsymbol{z}}_0(\boldsymbol{z}_t|\boldsymbol{y}_1, \phi_1) + \sqrt{1 - \bar{\alpha}_{t-1} - \sigma_t^2} \cdot \nabla_{\boldsymbol{z}_t} \log p_t(\boldsymbol{z}_t|\boldsymbol{y}_1, \phi_1) \qquad (24)$$

$$\mu_2 = \sqrt{\bar{\alpha}_{t-1}}\hat{\boldsymbol{z}}_0(\boldsymbol{z}_t|\boldsymbol{y}_2, \phi_2) + \sqrt{1 - \bar{\alpha}_{t-1} - \sigma_t^2} \cdot \nabla_{\boldsymbol{z}_t} \log p_t(\boldsymbol{z}_t|\boldsymbol{y}_2, \phi_2),$$

which defines a view-consistent diffusion model, whose score function is a weighted average of two Zero123 models:

$$\nabla_{\boldsymbol{z}_t} \log p_t(\boldsymbol{z}_t|\boldsymbol{y}_1, \phi_1, \boldsymbol{y}_2, \phi_2) = (1 - \gamma_t)\nabla_{\boldsymbol{z}_t} \log p_t(\boldsymbol{z}_t|\boldsymbol{y}_1, \phi_1) + \gamma_t \nabla_{\boldsymbol{z}_t} \log p_t(\boldsymbol{z}_t|\boldsymbol{y}_2, \phi_2),$$

$$\gamma_t = \frac{\sigma_t^2}{2\sigma_t^2 + \nu_t^2}. \qquad (25)$$

This derivation can be readily extended to scenarios with multiple unposed images.

**Annealing of View-consistent Diffusion.**    An annealing strategy is essential for the view-consistent diffusion process in pose-free 3D reconstruction due to initial inaccuracies in estimated camera poses. In a two-image-based 3D reconstruction problem, for instance, model errors are quantified by $\nu_t$ in Eq. 25. This error term starts large and gradually decreases to zero as the camera pose $\phi_2$ becomes increasingly accurate. Consequently, $\gamma_t$ progressively increases from nearly 0 to 0.5. During early stages, the diffusion process primarily relies on a single reference view $\boldsymbol{y}_1$, and the intermediate generative images are utilized to calibrate camera poses of other images. As camera poses gain accuracy, the other images exert growing influence on the diffusion process, ultimately achieving view-consistent results.

### D.3    M-STEP

The M-step estimates unknown camera poses by aligning unposed images to synthetic and reference views:

$$\phi_2 = \arg\min_{\phi} \mathbb{E}_{\hat{\boldsymbol{x}}_0} \left[ \lambda \|\boldsymbol{z}_t(\boldsymbol{y}_2, \phi_2) - \boldsymbol{z}_t(\hat{\boldsymbol{x}}_0, \boldsymbol{0})\|_2^2 + \delta\|\boldsymbol{z}_t(\boldsymbol{y}_2, \phi_2) - \boldsymbol{z}_t(\boldsymbol{y}_1, \phi_1)\|_2^2 \right], \quad (26)$$

where $\hat{\boldsymbol{x}}_0$ are the posterior samples from the E-step, and $\lambda$ and $\delta$ balance the calibration loss on the synthetic and reference images, respectively. $\boldsymbol{z}_t(\cdot, \cdot)$ is the time-dependent latent variable representing semantic information of the transformed input image. As suggested by Eq. 12, this optimization problem can be efficiently solved using gradient-based optimization. We dynamically adjust the ratio of the two balancing factors, $\lambda/\delta$, throughout the diffusion process. In early stages, we set $\lambda/\delta$ to a small value, primarily relying on the reference image for pose calibration. As the synthetic image becomes more realistic during the diffusion process, we gradually increase $\lambda/\delta$ until it converges to 1, balancing the influence of both synthetic and reference images in the final stages.

Table 3: Numerical results of 3D models generated using single (Zero123) and dual (ours) inputs. The measurements for 3D metrics (CD and VolumeIoU) are computed across 10 cases, while 2D metrics (PSNR, SSIM and LPIPS) were evaluated by generating 12 new viewpoint images (at 30-degree intervals) for each case, utilizing the same inputs as for 3D reconstruction.

| Methods | PSNR↑ | SSIM↑ | LPIPS↓ | CD↓ | VolumeIoU↑ |
|---------|-------|-------|--------|-----|------------|
| zero123 | 12.76 | 0.769 | 0.249 | 0.115 | 0.470 |
| Ours | **14.19** | **0.800** | **0.191** | **0.105** | **0.670** |

### D.4    3D RECONSTRUCTION FROM 2D NOVEL VIEWS

Previous sections primarily focused on synthesizing novel views from unposed input images, however, this technique can be extended to render 3D objects by generating multiple random novel views. Following the 3D reconstruction process described in One-2-3-45(Liu et al., 2024), we input these synthetic images and their corresponding poses into an SDF-based neural surface reconstruction module to achieve $360°$ mesh reconstruction. Fig. 8 and Table 3 demonstrate the results of this process, showing 3D reconstructions of both textured and textureless meshes derived from two unposed input images.

### D.5    MORE INPUT VIEWS YIELD BETTER RECONSTRUCTION

In Sec. D.2, Sec. D.3 and Sec. D.4, we primarily explore 3D reconstruction based on images from two unposed views. This approach is readily adaptable to scenarios involving an arbitrary number of views. We evaluate the quality of 3D reconstructions using one, two, and three unposed images, as shown in Fig. 9. The results indicate that adding more views significantly improves the fidelity of the 3D reconstruction. The LatentDEM framework facilitates consistent 3D reconstruction across different images. Specifically, 3D reconstruction from a single view (equivalent to Zero123) results in a hollow reconstruction, whereas incorporating additional views progressively yields more realistic 3D models.

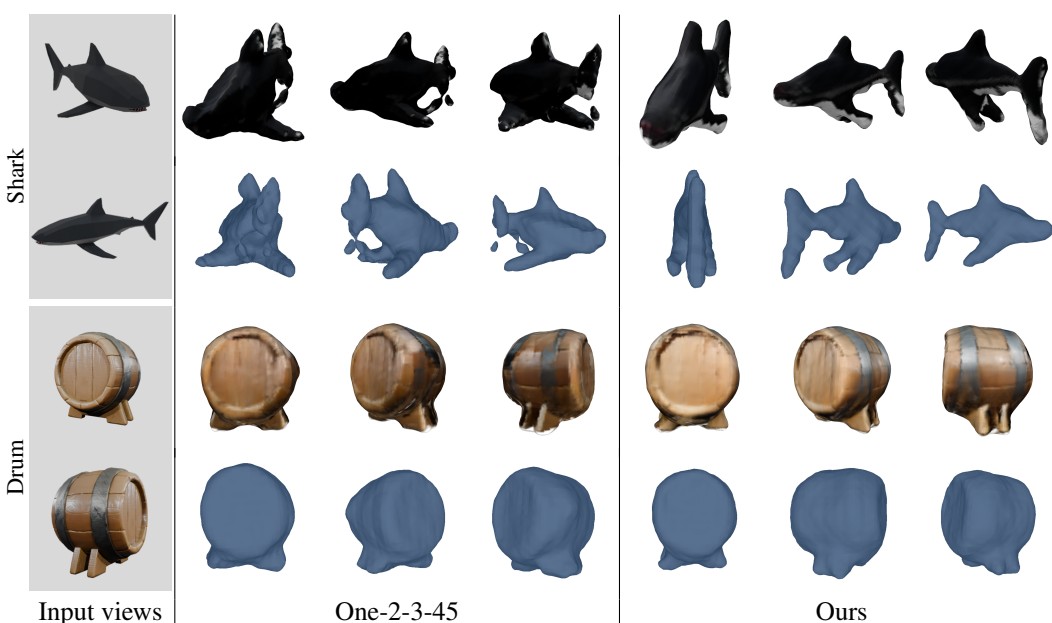

Figure 8: **3D mesh reconstruction with One-2-3-45 (Liu et al., 2024).** We compare our method's performance on 3D mesh reconstruction with One-2-3-45. Both texture and textureless meshes are shown. The baseline sometimes fails to recover fine details as they can only take one input view, while our method shows better mesh reconstruction with two input views.

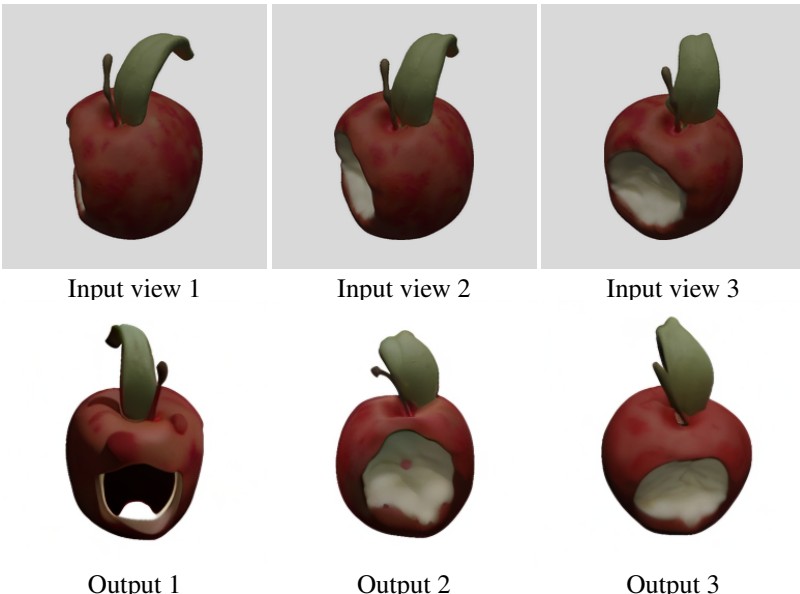

Figure 9: **3D reconstruction of an apple using different numbers of unposed images.** We evaluate our method's performance with varying numbers of input sparse views. **Left**: One view (Output 1 is from Input view 1 which is 1equivalent to Zero123) generates an unrealistic model that is hollow inside. **Middle**: Output 2 is from Input view 1 & 2. Two views improve results but still exhibit hallucinations in the 3D geometry of the bitten apple, *e.g.*the red dot in the middle of the bitten part. **Right**: Output 3 is from all three input views. Three views successfully recover all details.

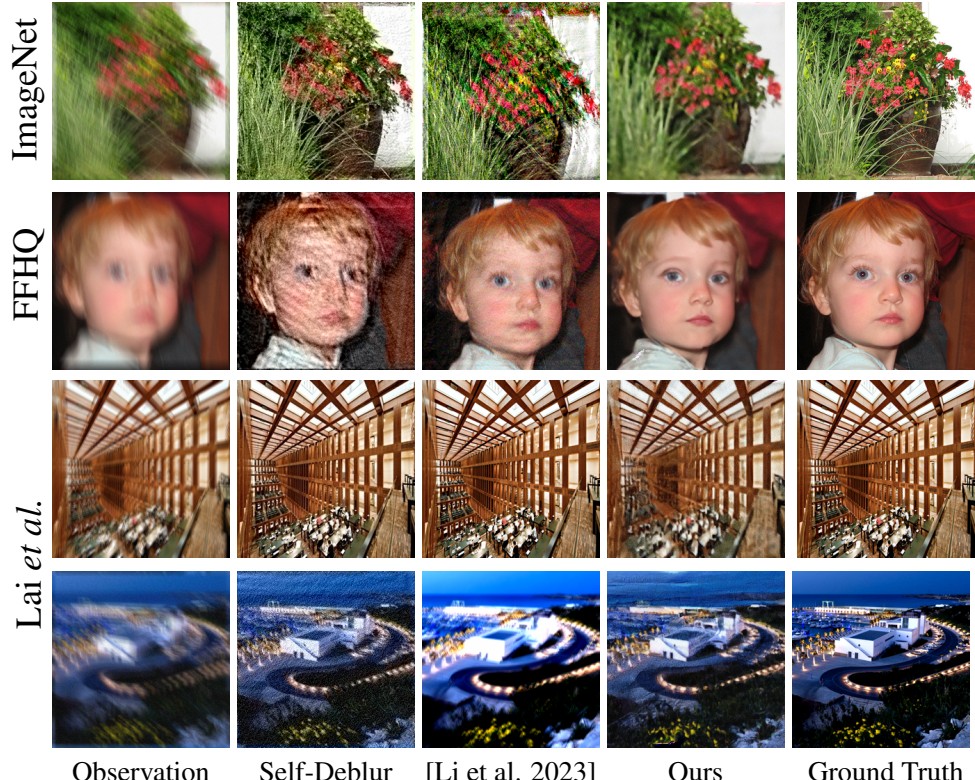

Observation     Self-Deblur     [Li et al. 2023]     Ours     Ground Truth

Figure 10: Comparison of blind motion deblurring results between LatentDEM, Self-Deblur , and the current SOTA self-supervised method Li et al. (2023) on various datasets (ImageNet, FFHQ, and Lai et al. (2016)).

Table 4: Comparison with SOTA self-supervised method Li et al. (2023) on ImageNet and FFHQ.

| Method | ImageNet | | | FFHQ | | |
|---|---|---|---|---|---|---|
| | PSNR↑ | SSIM↑ | LPIPS↓ | PSNR↑ | SSIM↑ | LPIPS↓ |
| Self-Deblur | 16.74 | 0.232 | 0.493 | 18.84 | 0.328 | 0.493 |
| Li et al. (2023) | 17.94 | 0.315 | 0.381 | 20.18 | 0.4 | 0.338 |
| Ours | 19.35 | 0.496 | 0.256 | 22.65 | 0.653 | 0.167 |

## E  MORE ABLATIONS AND EXPERIMENTAL RESULTS

**Comparison with SOTA Self-supervised Deblurring Method on Benchmark Dataset.**  We additionally compared LatentDEM with the current SOTA self-supervised deblurring method, specifically Li et al. (2023), using datasets from both the plug-and-play deblurring community (ImageNet,

Table 5: Average PSNR on Lai et al. (2016) dataset, compared with SOTA supervised learning method Li et al. (2022) and self-supervised method Li et al. (2023) across various categories.

| Method | Saturated | People | Natural | Text | Manmade | Average |
|---|---|---|---|---|---|---|
| Li et al. (2022) | 16.73 | 24.23 | 20.59 | 17.45 | 17.28 | 19.25 |
| Li et al. (2023), Reported | 17.21 | 31.02 | 26.00 | 25.46 | 23.06 | 24.55 |
| Li et al. (2023), ID Kernel | 12.96 | 17.02 | 22.72 | 12.98 | 14.93 | 16.12 |
| Li et al. (2023), OOD Kernel | 11.53 | 13.80 | 14.62 | 10.21 | 11.88 | 12.41 |
| Ours | 18.29 | 23.39 | 23.99 | 16.57 | 20.06 | 20.46 |

FFHQ) and the self-supervised deblurring benchmark (Lai et al. (2016)). The results are in Table 4, 5 and Fig. 10. In ImageNet and FFHQ datasets, we found the new baseline significantly improves over Self-Deblur. However, LatentDEM still outperforms it.

We further tested latentDEM across various categories from the Lai et al. (2016) dataset. Results are shown in Table 5, *Reported* denotes results in the original paper, *ID kernel* and *OOD kernel* denote the results we tested using their officially-released code, respectively using 4 kernels inside the dataset and other similar random kernels not included in the dataset. Our method outperformed SOTA supervised learning method Li et al. (2022) and self-supervised method Li et al. (2023) across various categories, though it still lags behind Li et al. (2023) in natural observations when kernels are ID, which indicates Li et al. (2023)'s sensitivity to the blur kernels. We want to emphasize that Li et al. (2023) typically requires different neural network architectures (e.g., channels of convolutional kernels), training epochs, and hyperparameters for different image categories to achieve optimal performance. This tuning process is very empirical, and we have tried our best to conduct the comparison fairly. In contrast, LatentDEM uses a pre-trained diffusion model, which requires minimal tuning and achieves decent results across most images.

**LDMVQ-4 v.s. Stable Diffusion-V1.5.** We evaluate LatentDEM's performance on 2D blind deblurring using two widely adopted foundational latent diffusion models: LDMVQ-4 and Stable Diffusion V1.5 Rombach et al. (2022). Fig. 12 presents the results of this comparison. Both models achieve satisfactory reconstruction outcomes, demonstrating LatentDEM's ability to generalize across various LDM priors when solving blind inverse problems. Notably, LatentDEM exhibits superior performance with Stable Diffusion compared to LDMVQ-4. This difference can be attributed to Stable Diffusion's more recent release and its reputation as a more advanced diffusion model.

**Vanilla EM v.s. Diffusion EM.** Traditional EM algorithms perform both E-step and M-step until convergence at each iteration before alternating. This approach guarantees a local optimal solution, as demonstrated by numerous EM studies. However, in the context of diffusion posterior sampling, which involves multiple reverse diffusion steps, this paradigm proves inefficient and computationally expensive.

Our implementation of the EMDiffusion algorithm for blind inverse tasks deviates from this conventional approach. Instead of waiting for the diffusion process to converge (typically 1,000 reverse steps with a DDIM scheduler), we perform model parameter estimation (M-step) after each single diffusion reverse step, except during the initial stage where we employ the skip gradient technique, as detailed in Sec. 4.2. Table 6 compares the performance of vanilla EM and our method on the blind deblurring task. Our approach, LatentDEM, requires only 1.5 minutes compared to vanilla EM's 120 minutes, while achieving superior reconstruction quality and kernel estimation.

Besides, we've tested running a second iteration of the full diffusion process once the forward model parameters are determined. Specifically, we first used our original setting to estimate the blur kernel, then fixed the kernel and ran another 1000 diffusion steps (equivalent to 1000 E-steps only) from Gaussian noise. As reported in Fig. 11, the quality of reconstructed images from the second iteration is comparable to the original single-iteration implementation. This is likely because our annealing techniques help avoid overfitting to wrong images with the guidance of inaccurate blur kernels at early diffusion stages, so the second iteration slightly helps but will not significantly improve results.

These results demonstrate that performing the M-step after each reverse step is both effective and efficient for blind inverse tasks. Moreover, this strategy offers improved escape from local minima and convergence to better solutions compared to vanilla EM, which completes the entire diffusion reverse process in each EM iteration.

Table 6: Vanilla EM v.s. Diffusion EM in blind deblurring.

| Method | Image | | | Kernel | | Time |
|---|---|---|---|---|---|---|
| | PSNR↑ | SSIM↑ | LPIPS↓ | MSE↓ | MNC↑ | Minute↓ |
| Vanilla EM | 20.43 | 0.561 | 0.419 | 0.024 | 0.124 | 120 |
| LatentDEM | 22.23 | 0.695 | 0.183 | 0.023 | 0.502 | 1.5 |

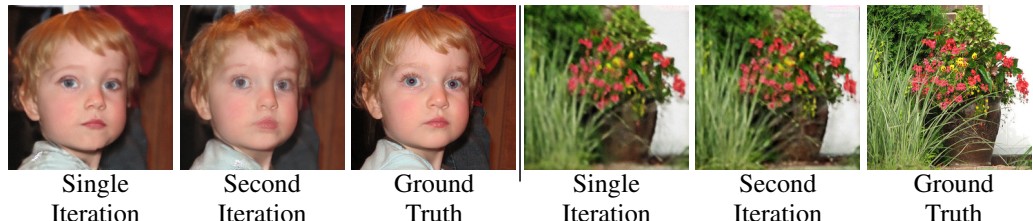

| Single Iteration | Second Iteration | Ground Truth | Single Iteration | Second Iteration | Ground Truth |

Figure 11: Comparison of blind motion deblurring results between the original single-iteration and suggested second-iteration diffusion. The quality of reconstructed images from the second iteration is comparable to the single-iteration results.

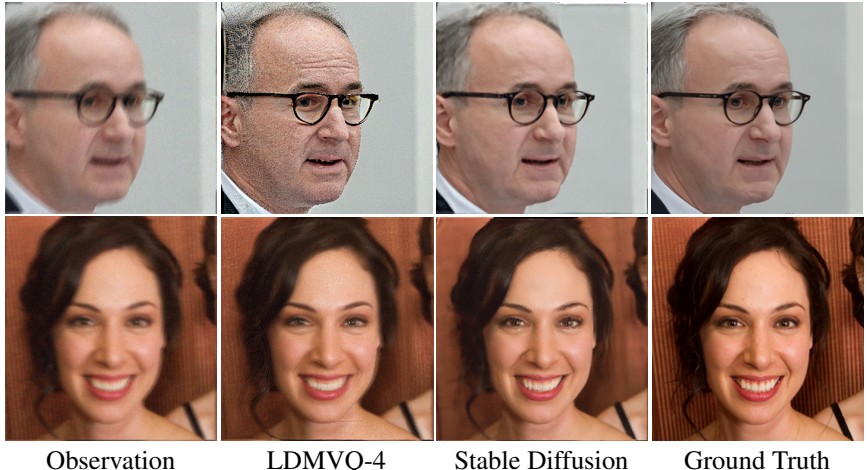

| Observation | LDMVQ-4 | Stable Diffusion | Ground Truth |

Figure 12: **LDMVQ-4 v.s. Stable Diffusion-V1.5.** Stable Diffusion V1.5 generates results with more detailed textures due to its more powerful priors.

**Additional Results of Pose-free Sparse-view Novel-view Synthesis.** Figure 13 presents additional results of novel-view synthesis using LatentDEM.

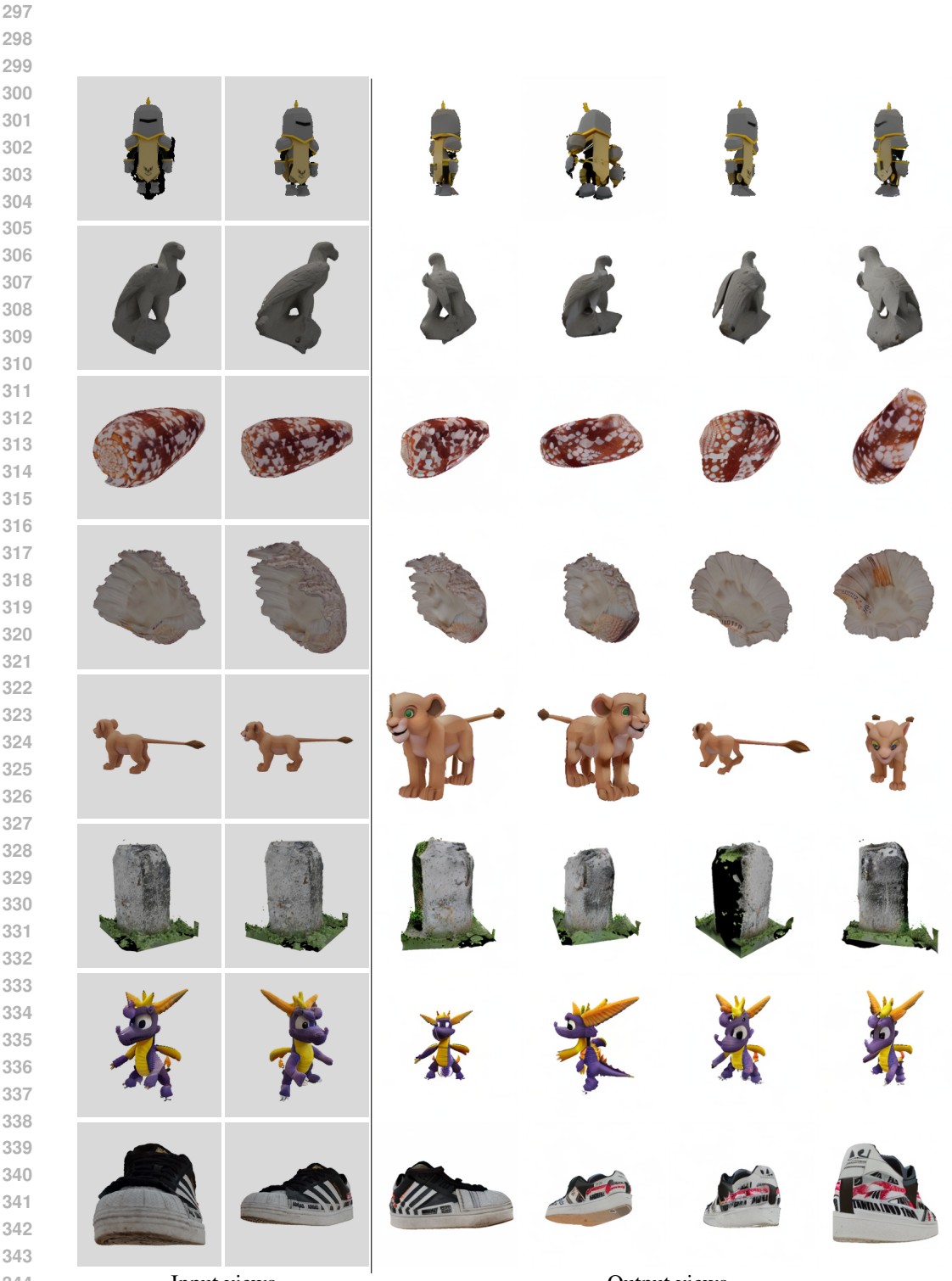

Input views        Output views

Figure 13: **Pose-free sparse-view novel-view synthesis.**

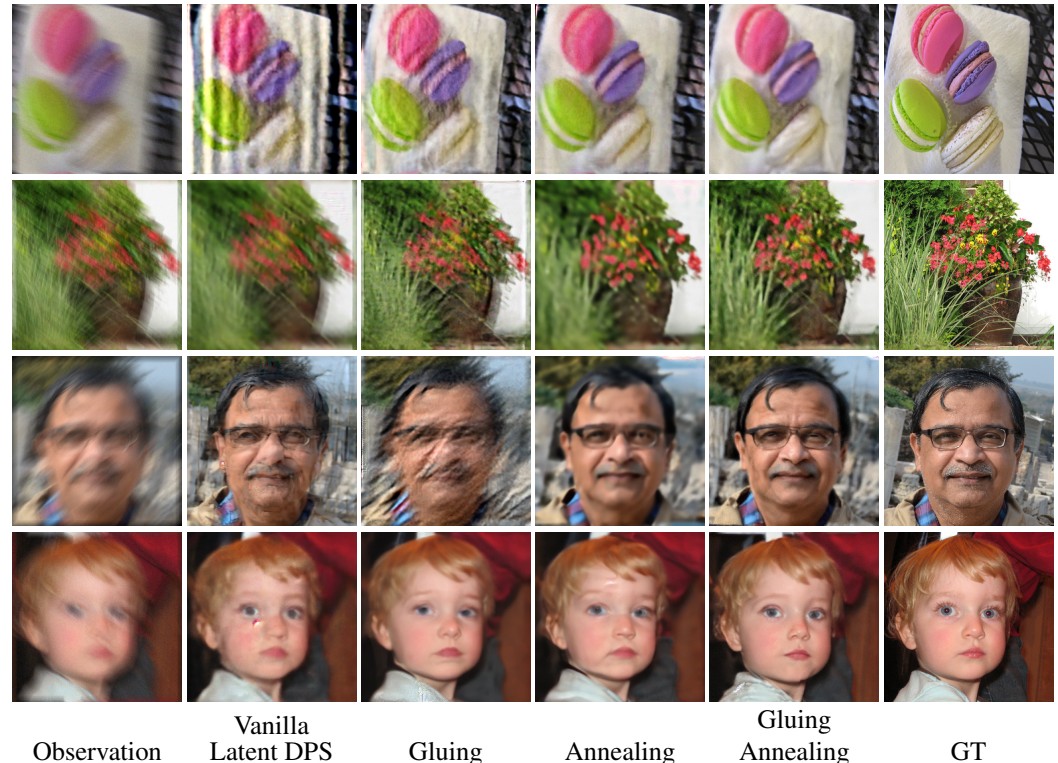

|                | Vanilla
Latent DPS | Gluing | Annealing | Gluing
Annealing | GT |

Observation    Vanilla Latent DPS    Gluing    Annealing    Gluing Annealing    GT

Figure 14: **The effect of annealing consistency technique and gluing term.** We find that both the gluing term and annealing consistency technique yield better results, while combining them achieves the best result.

## F    EXPERIMENTS FOR REVIEWERS

**Role of "Gluing" regularization vs. Annealing Consistency**    Figure 14 shows the isolation experiments to look into the effectiveness of the annealing consistency technique and the gluing term.

**FastEM Reproduce**    We provide additional results based on small kernels in Figure 15 to prove we have correctly reproduced the FastEM.

**Compare with GibbsDDRM**    We show the comparison results with an extra baseline GibbsD-DRM Murata et al. (2023) in Figure 16

**Divergence Results**

- HQS times: Figure 17 show some results with different HQS times and some will cause the results to diverge. Here HQS times means how many HQS optimizations we perform in one single M step.
- Annealing schedule: Figure 18 show some results with different annealing schedules and we can basically achieve good performances.
    - Annealing 1: $\zeta_t$ anneals linearly from 10 at $t = 1000$ to 1 at $t = 800$ and then holds

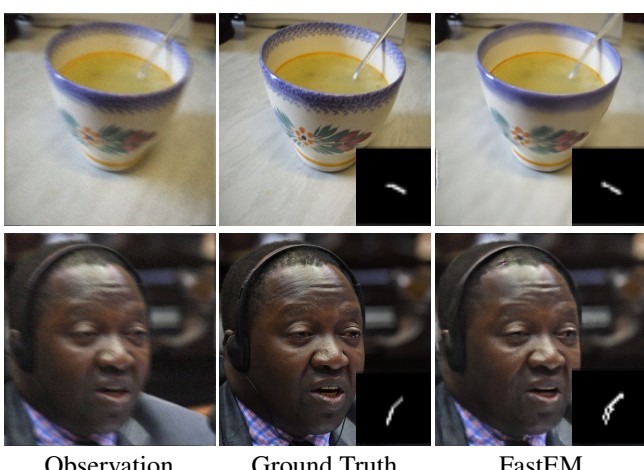

Observation          Ground Truth          FastEM

Figure 15: **FastEM Reproduce results**. When the kernel is rather simple and small, FastEM can produce good results.

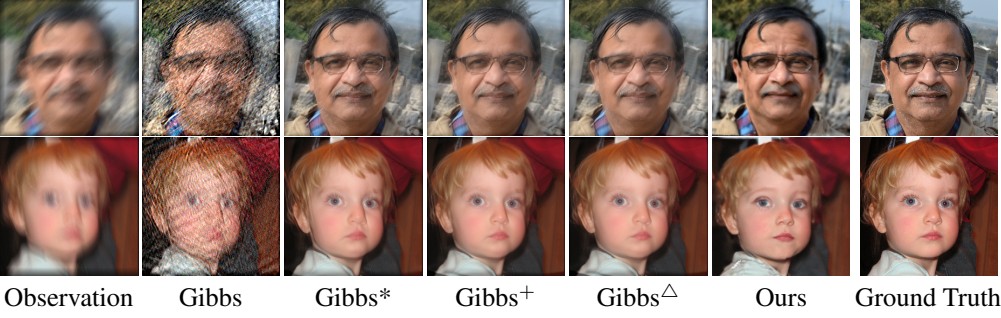

Observation    Gibbs    Gibbs*    Gibbs$^+$    Gibbs$^\triangle$    Ours    Ground Truth

Figure 16: **Comparison with GibbsDDRM**. Our method still outperforms GibbsDDRM with their official implementation and checkpoint, even though we have finely tuned the hyperparameters (Gibbs*: $\sigma$=5 for normalization, Gibbs$^+$: iterHupdate $= 10$ for updating kernels and Gibbs$^\triangle$: $\lambda = 500$ for weighting Laplace prior).

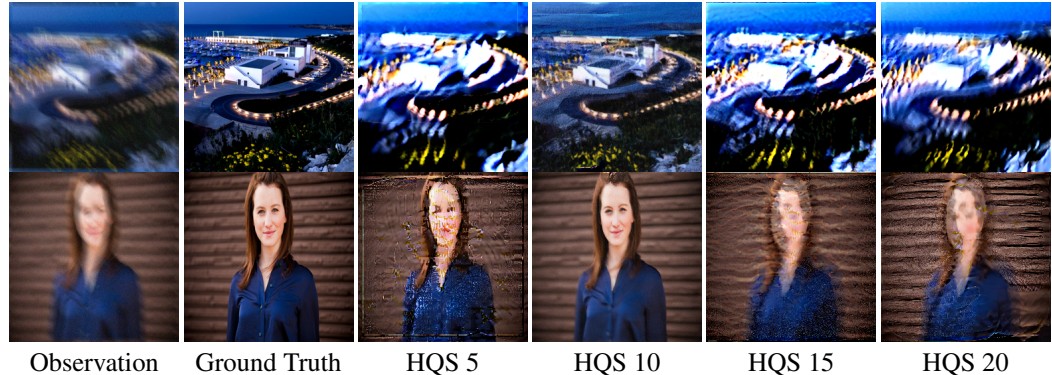

|   |   |   |   |   |   |
|---|---|---|---|---|---|
| Observation | Ground Truth | HQS 5 | HQS 10 | HQS 15 | HQS 20 |

Figure 17: **HQS times cause divergence.** If the HQS times are not chosen carefully, it will lead to divergence results.

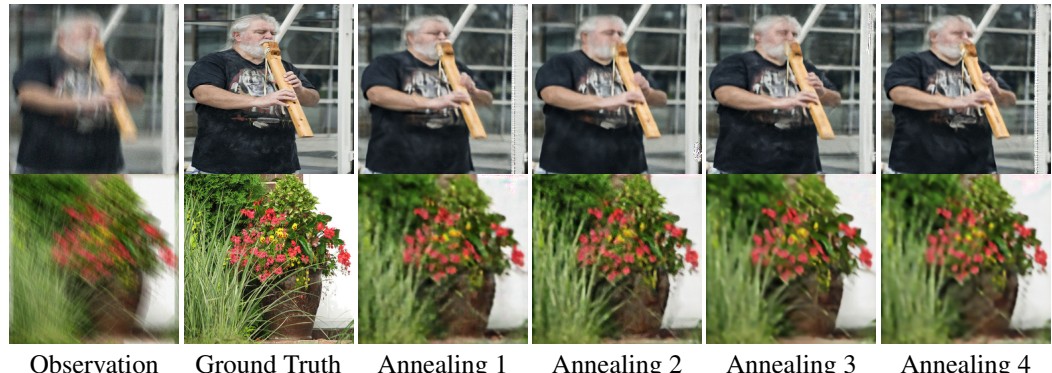

|   |   |   |   |   |   |
|---|---|---|---|---|---|
| Observation | Ground Truth | Annealing 1 | Annealing 2 | Annealing 3 | Annealing 4 |

Figure 18: **Different annealing schedules cause degradation.** Different annealing schedules won't lead to divergence but will cause slightly degradation compared with the optimal results.

- Annealing 2: $\zeta_t$ anneals linearly from 10 at $t = 1000$ to 1 at $t = 600$ and then holds
- Annealing 3: $\zeta_t$ anneals linearly from 10 at $t = 1000$ to 1 at $t = 200$ and then holds
- Annealing 4: $\zeta_t$ anneals linearly from 10 at $t = 1000$ to 0.8 at $t = 600$ and then holds

## G  DERIVATION OF EXPECTATION-MAXIMIZATION ALGORITHM

The Expectation-Maximization (EM) algorithm Dempster et al. (1977); Murphy (2023) is an iterative optimization method used to estimate the parameters $\phi$ of a statistical model that involves underlying (latent) variables $x$, given observations $y$. Below, we provide a step-by-step derivation based on maximizing the marginal log-likelihood $\log p_\phi(y)$.

### STEP 1: MARGINAL LOG-LIKELIHOOD

The goal of EM is to maximize:

$$\log p_\phi(y) = \log \int p_\phi(y, x)\, dx. \tag{27}$$

### STEP 2: INTRODUCING AN AUXILIARY DISTRIBUTION $q(x|y)$

Using an auxiliary distribution $q(x|y)$, the marginal log-likelihood can be rewritten as:

$$\log p_\phi(y) = \log \int q(x|y) \frac{p_\phi(y, x)}{q(x|y)}\, dx. \tag{28}$$

### STEP 3: APPLYING JENSEN'S INEQUALITY

By applying Jensen's inequality, we obtain a lower bound on $\log p_\phi(y)$:

$$\log p_\phi(y) \geq \int q(x|y) \log \frac{p_\phi(y, x)}{q(x|y)}\, dx. \tag{29}$$

### STEP 4: SIMPLIFYING THE LOWER BOUND

The lower bound becomes:

$$\int q(x|y) \log \frac{p_\phi(y, x)}{q(x|y)}\, dx = \int q(x|y) \left[\log p_\phi(y, x) - \log q(x|y)\right]\, dx. \tag{30}$$

Decomposing $p_\phi(y, x)$ into $p_\phi(y|x)$ and $p(x)$, we have:

$$\int q(x|y) \log p_\phi(y, x)\, dx = \int q(x|y) \left[\log p_\phi(y|x) + \log p(x)\right]\, dx. \tag{31}$$

Rewriting the lower bound :

$$L(q, \phi) = \int q(x|y) \left[\log p_\phi(y|x) + \log p(x) - \log q(x|y)\right]\, dx. \tag{32}$$

Therefore, maximizing $\mathcal{L}(q, \phi)$ is equivalent to maximizing $\log p_\phi(y)$.

