# OpenReview forum: "Blind Inversion using Latent Diffusion Priors"
_ICLR.cc/2025/Conference — Submitted to ICLR 2025_

### Official Review · Reviewer_rjXb · 2024-10-21

**Soundness:** 2
**Presentation:** 2
**Contribution:** 2
**Rating:** 6
**Confidence:** 4

**Summary:**

The paper presents a new diffusion-based model for blind inverse problems, which alternates between a diffusion step with a fixed parameter (eg a fixed blur kernel), and a step estimating the kernel given an estimate of the image. The paper motivates this algorithm as an instance of the expectation-maximization (EM) framework. In particular, the approach leverages powerful latent diffusion models such as Stable Diffusion v1.5, and performs the bulk of the diffusion in the latent space.

The method is demonstrated in blind image deblurring and sparse view 3D reconstruction, obtaining a better performance in terms of both distortion and perceptual quality than the competing baselines.

**Strengths:**

- introduces an approach that leverages pretrained encoder/decoders which reduces the computational burden of working in the pixel space.
- good experimental results in blind deblurring and sparse view 3D reconstruction.

**Weaknesses:**

- I believe the link to the EM framework is not sufficiently well founded: the MAP kernel estimation step uses \hat{x_0} instead of a sample from the posterior p(x|y, \phi). It is not clear to me what is the role of q(x|y) in equation 5, and more generally the equation does not add much to the explanation in my opinion.
- The experimental setup is not sufficiently well described in the main text, i) raises doubts about testing on a training set (see related questions below) and ii) the papers seems to use circular padding which is not a very realistic setting.
- The presentation of the paper could be improved:
    - the citation style is incorrect in many parts of the paper (use citep{} for references that are not part of a sentence)
    - the mathematical notation is not consistent: equation 11 uses Z uppercase for denoting a vector, whereas the rest uses lowercase, the letter 'z' is used for both latent space vectors and auxiliary variables in 11, the letter D is used for a denoiser and a decoder. What is the  'F^2' operator in equation 11? Acronyms are defined multiple times throughout the text (eg. EM). The method Zero123 is not followed by a reference when introduced on page 7.

**Questions:**

- What is the denoiser used for estimating the blur kernels? how was that denoiser trained?
- Is there an overlap between training kernels and test kernels? How were the test kernels generated/chosen?
- Was the Stable Diffusion model was trained on the FFHQ and ImageNet images used for testing the models?

---

> ### Author Response · Authors · 2024-11-21
>
> We thank the reviewer for their constructive suggestions. We are pleased that the reviewer recognizes the "good experimental results in blind deblurring and sparse view 3D reconstruction." Below, we address the raised concerns in detail:
>
> ---
>
> 1. **Mathematical Formula in EM Framework**
>
> We agree with the reviewer that kernel estimation (M-step) should utilize posterior samples from $p(x|y)$ instead of a point estimate. This is exactly what we implement in this work. Leveraging Tweedie’s formula[1], $\hat{x}_0$, approximates posterior samples based on intermediate reverse diffusion states $(x_t)$:
>
> $x_0 \sim p(x|y) \rightarrow \hat{x}_0 \sim p(x_0|x_t)p(x_t|y)$,
>
> where $p(x_0|x_t)$ is defined by $\frac{1}{\sqrt{\bar{\alpha}(t)}}[x_t+(1-\bar{\alpha}(t))s_\theta(x_t,t)]$. Note that since $x_t$ is drawn from a posterior distribution conditioned on the observation $y$, $\hat{x}_0$ also represents a set of posterior samples, allowing us to retain the core principle of the expectation-maximization.
>
> This approximation is utilized because the forward models are inaccurate in the initial stages of the blind inversion problem. Executing the entire reverse diffusion process (~1000 steps) is computationally intensive and, as shown in Appendix E and Table 6 of the paper, often worsens reconstruction by incorporating excessive model errors. Instead, leveraging approximate samples from intermediate states for fast model correction yields better performance.
>
> Additionally, $q(x|y)$ in Equation 5 is an auxiliary distribution that we introduced to help the derivation of the EM algorithm, as done in many prior works [2,3]. The EM algorithm optimizes L(q, $\phi$) over this auxiliary distribution q(x|y) and the model parameter $\phi$ . According to Jensen’s inequality, the upper bound of this objective function is, $\log p_\phi (y)$, which is achieved only when $q(x|y)$ matches the true posterior $p_\phi(x∣y)$. Hence, the E-step maximizes the objective function by conducting posterior sampling of $p_\phi(x∣y)$. We have revised Equation 5 and accompanying explanations to make this clearer.
>
> ---
>
> 2. **Details about the Experimental Setup**
>
>     1) **Denoiser**. The denoiser is a DnCNN [4] with 5 blocks and 32 channels. We used 60,000 motion blur kernels [5] for training, each combined with an additional channel for the noise level map. The network was trained for 5,000 epochs using MSE loss with a learning rate of 1e-5. Training was conducted on an NVIDIA A100 GPU, taking approximately seven hours. The information is already provided in Appendix C.2.
>
>     2) **Training kernels and test kernels**. All blur kernels were generated via stochastic walks in 2D space [5], following the BlindDPS setting. Training and test kernels are non-overlapping. We will explicitly state this in the revision.
>
>     3) **Circular padding**. We acknowledge that circular padding may seem unrealistic for image processing. However, it was used in both baselines, BlindDPS and FastEM, so we leverage the same settings to ensure fair comparisons. We will clarify this in the revised manuscript.
>
>     4） **Stable Diffusion**.  Stable Diffusion is trained on the LAION 5B dataset, while the testing datasets are ImageNet and FFHQ. Although all datasets are collected from the Internet, prior work [6] has shown that they belong to different data distributions. Therefore, the computational imaging community typically considers them as non-overlapping and widely uses ImageNet and FFHQ for evaluation.
>
> ---
>
> 3. **Presentation**
> Thank you for highlighting these inconsistencies. We apologize for the oversight and appreciate your attention to detail. We have corrected these issues and made the manuscript more consistent:
>
>     1) **Citation styles**: Citation styles have been corrected in the updated manuscript.
>     2) **Equation 11**: You are correct that it should not contain F^2. The correct HQS algorithm has been updated in our revision.
>
> We have also addressed other typos and missing citations to improve clarity and consistency throughout the paper.
>
> ---
>
> We thank the reviewer once again for the insightful feedback and hope our responses address all concerns. We are happy to provide further clarifications in additional discussions. We kindly hope the reviewer could consider raising the score if satisfied with our responses. Thank you!
>
> ---
>
> **Reference**
>
> [1] Bradley Efron. Tweedies Formula and Selection Bias. 2011.
>
> [2] Jaemo Sung, et al. Latent-Space Variational Bayes, PAMI 2008.
>
> [3]  MATTHEW, et al. The Variational Bayesian EM Algorithm for Incomplete Data: with Application to Scoring Graphical Model Structures. Bayesian Statistics 2000.
>
> [4] Kai Zhang, et al. Plug-and-play Image Restoration With Deep Denoiser Prior, TPAMI 2021.
>
> [5] Fabien Gavant, et al. A physiological camera shake model for image stabilization systems, SENSORS 2011.
>
> [6] Litu Rout, et al. Solving Linear Inverse Problems Provably via Posterior Sampling with Latent Diffusion Models, NeurIPS 2023.

---

> > ### Comment · Reviewer_rjXb · 2024-11-27
> >
> > Thank you for the answers to my questions and concerns.
> >
> > I believe the notation of the paper has improved, and some experimental details are more clear now. I still share some concerns as with other reviews whether the algorithm can be strictly understood as EM, or some kind of well-engineered alternated estimation/optimization procedure. I will raise my score by one point since it reflects my final evaluation.
> >
> > btw, I must say that I found the answer
> >
> > > Although all datasets are collected from the Internet, prior work [6] has shown that they belong to different data distributions. Therefore, the computational imaging community typically considers them as non-overlapping and widely uses ImageNet and FFHQ for evaluation.
> >
> > not very satisfying - I was expecting a more quantitative proof that there is no overlap between datasets or at least an explanation from [6] of why this is the case. Who is the "computational imaging community"?

---

> > > ### Author Response · Authors · 2024-11-27
> > >
> > > Thank you for your thoughtful feedback. We appreciate your recognition of the improvements made to the notation and experimental details in the manuscript.
> > >
> > > **Regarding whether the algorithm can be strictly classified as an EM method**, we believe our approach is more like the EM algorithm rather than the alternative optimization, because our E-step conducts diffusion posterior sampling (DPS) rather than simply gradient-based point estimation. We will further clarify the relationship between our approach and EM in the revision. In addition, we will try to make our algorithm perform more like the standard EM algorithm by sampling more times in the E-step, and report the new experiments in our final revisions.
> > >
> > > **Regarding the dataset overlap issue** LAION-5B, the training set of Stable Diffusion v1.5, is a vast dataset comprising over 5.85 billion image-text pairs collected from the internet.  It was not specifically curated to include or exclude ImageNet content. Given the dataset's vastness and diversity, its overlap with ImageNet is minimal and negligible. Consequently, PSLD [1] utilized ImageNet as a test set for the Stable Diffusion model trained on LAION-5. Our paper adopted the same settings. Some analysis papers [2] also reveal that ImageNet and LAION-5B are statistically very different.
> > >
> > > Thank you again for your constructive comments and your decision to raise your score. We hope these additional clarifications will strengthen the manuscript and address your remaining concerns.
> > >
> > > **Reference**
> > >
> > > [1] Litu Rout, et al. Solving Linear Inverse Problems Provably via Posterior Sampling with Latent Diffusion Models, NeurIPS 2023.
> > >
> > > [2] Ali Shirali, et al. What Makes ImageNet Look Unlike LAION. arXiv 2024.

---

### Official Review · Reviewer_yUwp · 2024-10-29

**Soundness:** 3
**Presentation:** 4
**Contribution:** 2
**Rating:** 6
**Confidence:** 4

**Summary:**

This paper introduces LatentDEM, a novel method for addressing blind inverse problems through latent diffusion models. The proposed method leverages an Expectation-Maximization (EM) framework: in the expectation step, a clean image is recovered using a latent diffusion prior, and in the maximization step, the forward model is estimated based on the recovered clean image. The authors demonstrate the method’s application to blind deblurring and 3D pose-free sparse-view reconstruction.

**Strengths:**

The paper is well-organized and clearly communicates its results.

The authors provide extensive experimental results, comparing their method to prominent existing methods for blind inverse problems, such as BlindDPS and FastEM.

They provide an ablation study examining the influence of gradient estimation skipping and annealing consistency on performance.

**Weaknesses:**

1. A primary concern is the paper’s original contribution. The methodology appears to combine elements previously explored in the literature. For instance, latent diffusion models for inverse problems have been investigated in [1], while Expectation-Maximization approaches for blind inverse problems have been studied in [2]. Simply substituting a latent diffusion model as the prior does not, in itself, seem to constitute a novel contribution. Explicitly listing the contributions could improve clarity.

2. Another concern is the runtime of the proposed method. As seen in Table 2, the method appears to trade time for performance, with comparable results to BlindDPS but with similar computational demands. In other words, we would not gain any performance advantage compared to BlindDPS, if are limited by the runtime of the method.

3. The reported performance metrics for FastEM and BlindDPS appear inconsistent with previous results. For instance, in the FastEM paper, performance on the FFHQ dataset is as follows: MPRNet: 19.52, BlindDPS: 24.05, and FastEM: 25.75, demonstrating comparability to DPS given the GT kernel. However, in this paper, the results are MPRNet: 21.60, BlindDPS: 22.58, and FastEM: 17.46. Could the authors provide an explanation for this performance discrepancy? Is the comparison of FastEM accurate? Can they reproduce FastEM’s results by strictly following its settings for a more direct comparison?

[1] Litu Rout, Negin Raoof, Giannis Daras, Constantine Caramanis, Alex Dimakis, and Sanjay Shakkottai. Solving linear inverse problems provably via posterior sampling with latent diffusion models. Advances in Neural Information Processing Systems, 36, 2024.

[2] Charles Laroche, Andrés Almansa, and Eva Coupete. Fast diffusion em: a diffusion model for blind inverse problems with application to deconvolution. In Proceedings of the IEEE/CVF Winter Conference on Applications of Computer Vision, pp. 5271–5281, 2024.

**Questions:**

Have the authors observed any instances of non-converging patterns in their algorithm? Specifically, does a particular choice of annealing or tunable parameters (Equations 11 or 12) lead to divergence?

---

> ### Author Response · Authors · 2024-11-21
>
> We thank the reviewer for the constructive suggestions and feedback. We are pleased that the reviewer finds the paper “well-organized” and acknowledges our “extensive experimental results” and comprehensive “ablation study.” Below, we provide detailed responses to the concerns raised:
>
> ---
>
> 1. **Novelty and contribution**
>
> The integration of latent diffusion models with the Expectation-Maximization (EM) framework for inverse problems is far from trivial, contrary to the reviewer’s assumption. Prior work on blind inversion has demonstrated that jointly optimizing model parameters and underlying images introduces severe non-convexity, making image reconstruction inherently unstable.
>
> The introduction of latent diffusion models further amplifies this challenge, because their encoder-decoder structure adds fragility and computational inefficiency due to the nonlinear mapping between pixel and latent spaces. To address these challenges, we introduced novel techniques such as: 1) "annealed consistency" to mitigate model errors and 2) "skip gradients" to accelerate computation.
>
> Our method not only outperforms all existing diffusion-based blind deblurring baselines but also, for the first time, demonstrates capability in solving challenging pose-free 3D reconstruction tasks. We will explicitly highlight these key contributions in the revised manuscript.
>
> ---
>
> 2. **Runtime compared to BlindDPS**
>
> We agree that when runtime is heavily constrained (as shown in Table 2), our method must trade reconstruction quality for inference speed. However, for scenarios that prioritize quality over speed—such as scientific imaging tasks like microscopy denoising or super-resolution—our method offers an alternative solution with better reconstruction accuracy.
>
> Additionally, BlindDPS relies on training task-specific priors, whereas our approach eliminates this requirement by utilizing a pre-trained Stable Diffusion model, significantly reducing the computational overhead in the training phase.
>
> ---
>
> 3. **Performance regarding FastEM and BlindDPS**
>
> The performance differences between our results and those reported in the FastEM paper are primarily due to the different motion blur kernel sizes. Larger kernels define harder blind inversion tasks, degrading reconstruction quality. Since FastEM does not specify the kernel size used in the paper, we tested a range of sizes and found that FastEM only performs well with small kernels but struggles significantly with larger, more complex ones.
>
> We present experimental results of FastEM with small kernels in Appendix F of the revision. We successfully replicated FastEM’s reported performance for small kernels but observed significant performance drops with larger kernels in Figure 3.
>
> Another evidence supporting our findings comes from the BlindDPS paper, where the reported PSNR of 22.24 aligns closely with the 22.58 reported in our paper but is significantly lower than the 24.05 reported by FastEM. This suggests that FastEM defined an easier task than the original BlindDPS paper but hides this critical information in the paper.
>
> Additional experiments supporting these claims have been included in the updated manuscript (Appendix. F). Our comparisons in the paper are fair and accurate.
>
> ---
>
> 4. **Non-converging patterns from tunable parameters**
>
> 1) Annealing parameters: We did not observe non-converging patterns with annealing. On the contrary, annealing consistently enhances performance by preventing reconstructions from overfitting to model errors during diffusion posterior sampling. As shown in Fig. 5, removing annealing parameters results in oscillatory artifacts but does not lead to divergence, annealing at least performs better than in this case. We also included additional results for further validation in the updated manuscript (Appendix F).
>
> 2) Optimization Hyperparameters: Equations 11 and 12 define iterative optimization procedures for model parameter estimation. If learning rates are too large or noise parameters deviate from true noise levels, the optimization can become unstable, producing non-converging patterns. This is a common challenge across all optimization algorithms but can be resolved with careful hyperparameter tuning.
>
> ---
>
> We thank the reviewer again for their insightful suggestions and hope our responses address all concerns. We are happy to provide further clarifications in additional discussions. We kindly hope the reviewer could consider raising the score if satisfied with our responses. Thank you!

---

> > ### Comment · Reviewer_yUwp · 2024-11-22
> > **Response to the authors**
> >
> > I thank the authors for their detailed rebuttal. I have carefully reviewed both the rebuttal and the other reviews. The clarification regarding the comparison with FastEM was helpful, and I believe the paper’s contributions—particularly the improvement over BlindDPS and its application to pose-free sparse-view 3D reconstruction—are valuable.
> >
> > That said, I find the following statement problematic:
> >
> >  > Our method not only outperforms all existing plug-and-play blind deblurring baselines ...
> >
> > Based on the results presented in [1], the PnP approach for blind inverse problems (deblurring) appears to outperform “SelfDeblur” by a large margin and also achieves better SSIM scores compared to BlindDPS. If the authors can provide additional justification for this claim, it would address this concern. Otherwise, I suggest revising or removing the statement to maintain accuracy and clarity.
> >
> > [1] Gan, Weijie, et al. "Block coordinate plug-and-play methods for blind inverse problems." Advances in Neural Information Processing Systems 36 (2024).

---

> > > ### Author Response · Authors · 2024-11-22
> > >
> > > We apologize for the confusion: we intended to emphasize our method’s superior performance over **diffusion-based baselines** rather than **Plug-and-Play baselines**. We have revised the statement to avoid further confusion, and are happy to include PnP-based baselines in the final version. Thank you for bringing this to our attention.

---

> > > > ### Comment · Reviewer_yUwp · 2024-11-27
> > > >
> > > > I thank the authors for the clarifications. I have adjusted my score due the clarification on the contributions and comparison with the baselines.

---

### Official Review · Reviewer_E53D · 2024-10-30

**Soundness:** 2
**Presentation:** 1
**Contribution:** 2
**Rating:** 5
**Confidence:** 4

**Summary:**

The paper proposed a method for solving blind restoration problems (examined mostly on blind deblurring) using a pretrained latent diffusion model (LDM). It is based on adding to the iterations of unconditional LDM sampling steps (which are sometimes skipped) where the the observation/forward model parameters (e.g., blur kernel) are updated and being used for guiding the diffusion (I do not see a clear relation to expectation maximization). The proposed method is quite heuristic and shares similar ideas with existing works that use guided LDM for non-blind restoration and works that use guided pixel-space DMs for blind restoration. In the reported experiments it outperforms alternative methods.

**Strengths:**

- As far as I know, this is the first work that uses a pretrained LDM for blind restoration.

- The proposed method appears to have empirical advantages over competitors.

**Weaknesses:**

- The contribution seems incremental in terms of the techniques being used and the lack of theoretical motivation and analysis of the proposed method.

- The presentation is not good in terms of explaining why the authors refer to the proposed method as an expectation maximization (EM) technique.

**Questions:**

- The presentation of EM in Section 3 differs from the plain EM formulation (as in Dempster et al., 1977 and textbooks).
For example, what are the unobserved/missing data? Where in the E-step do you compute (conditional) expectation of the log likelihood/posterior with respect to the missing data?
The authors may refer to some variational Bayes modification of EM but not to the actual EM. Make the discussion more clear and point to the reference that justifies your formulation.

- Consequently, the presentation of the method in Section 4 needs to be improved. The derivation of the method does not match the EM paradigm. It looks like some alternating optimization scheme that lacks formal connection to EM.

- Lack of theoretical analysis for the proposed modifications to the baseline alternating optimization approach.
For "Technique 1" there is some motivation in the appendix, but no formal mathematical argument that actually considers the effect of error in phi. The other modifications are quite heuristic.

- Regarding the experiments, you need to provide more details on the kernels, noise levels, etc., that you use in training and in testing.
Also, state in some table the hyperparameters that you use. Do you use the same hyperparameters for all the settings and for both 2D and 3D?

-  The reported results of the competitors for blind deblurring on FFHQ are lower than those reported in previous works  (Chung et al., 2022a), (Laroche et al., 2024) (e.g., worse kernel estimation for BlindDPS and huge deterioration for FastEM).
How can you explain it? Do you use the same setup as these works? If not, then explain why.

- Please compare also with GibbsDDRM:
Murata, Naoki, et al. "Gibbsddrm: A partially collapsed gibbs sampler for solving blind inverse problems with denoising diffusion restoration." ICML, 2023.

- Present also the results for the non-blind methods on which BlindDPS and FastEM, and your approach, are based. They can be interpreted as the upper bound on the results, as done in GibbsDDRM and FastEM.

---

> ### Author Response · Authors · 2024-11-21
> **Official Comment by Authors (Part 1)**
>
> We thank the reviewer for their constructive suggestions. We are pleased that the reviewer recognizes our work as "the first to use a pre-trained LDM for blind restoration." Below, we provide detailed explanations and clarifications for the concerns raised:
>
> ---
>
> 1. **Novelty and contribution**
>
> The integration of latent diffusion models (LDMs) with the Expectation-Maximization (EM) framework for inverse problems is far from trivial, contrary to the reviewer’s assumption. Prior work on blind inversion has shown that jointly optimizing model parameters and underlying images introduces severe non-convexity, making image reconstruction highly unstable.
>
> The inclusion of LDMs amplifies these challenges due to the encoder-decoder structure, which introduces additional fragility and computational inefficiency through the nonlinear mapping between pixel and latent spaces. To address these issues, we developed two techniques: 1) "annealed consistency" to mitigate model errors and 2) "skip gradients" to accelerate computation.
>
> For imaging applications, our method not only outperforms all existing diffusion-based blind deblurring baselines but also demonstrates, for the first time, capabilities in solving challenging non-linear, pose-free 3D reconstruction tasks. We will explicitly highlight these contributions in the revised manuscript.
>
> ---
>
> 2. **Expectation-Maximization algorithm and the derivation of our method**
>
> Our method does employ an EM framework, not an alternating optimization approach. In this framework, the true images ($x$) are the latent variables, and the goal is to estimate model parameters ($\phi$) from observations ($y$) without knowing $x$.
>
> In the E-step, we generate multiple posterior samples using diffusion posterior sampling. For simplicity and robustness, we approximate the posterior via Tweedie’s formula [1]:
>
> $x_0 \sim p(x|y) \rightarrow \hat{x}_0 \sim p(\hat{x}_0|x_t)p(x_t|y)$,
>
> where $p(x_0|x_t)$ is defined by $\frac{1}{\sqrt{\bar{\alpha}(t)}}[x_t+(1-\bar{\alpha}(t))s_\theta(x_t,t)]$, based on the intermediate state $(x_t)$. Thanks to the property of the EM algorithm, it has a theoretical guarantee of local convergence.
>
> The reviewer’s misunderstanding may stem from how we introduced $q(x|y)$ in Equation 5. $q(x|y)$ is an auxiliary distribution to aid the EM derivation, not a variational distribution. The EM algorithm optimizes $L(q, \phi)$ over q(x|y) and $\phi$. By Jensen’s inequality, the upper bound, $L(q, \phi) = \log p_\phi (y)$, is achieved only when $q(x|y)$ matches the true posterior $p_\phi(x∣y)$. Thus, our E-step conducts posterior sampling of $p_\phi(x∣y)$ (via stable diffusion), while the M-step maximizes the expected likelihood. We have revised Equation 5 and its accompanying explanation to clarify this point.
>
> ---
>
> 3. **Details about the Experimental Setup**
>
> 1) Imaging Task Parameters: For the blind deblurring task, the blur kernels are $64 \times 64$ pixels, generated using random walks. The training and testing kernels are non-overlapping. Additionally, the images are assumed to have additive Gaussian noise with $\sigma=0.02$.
>
> 2) Algorithm Hyperparameters: The hyperparameters differ between the 2D and 3D tasks. For clarity, these differences are outlined in the table below.
>
> | Task | Steps      | Annealing (E step)                                     | $\lambda / \gamma$ | M step                             |
> |------|------------|--------------------------------------------------------|----------------|------------------------------------|
> | 2d   | 1000       | Linear                                                 | 1e-5           | HQS optimization for 10 times     |
> | 3d   | 100 (ddim) | Measurement-loss based (refer to paper Eq. 25)          | 1e-5           | Gradient descent: Scale: 1        |
>
>
>
> ---
>
> **Reference**
> [1] Bradley Efron. Tweedies Formula and Selection Bias. 2011.

---

> > ### Comment · Reviewer_E53D · 2024-11-24
> >
> > Regarding "The reviewer’s misunderstanding...":
> >
> > There is no misunderstanding on my part regarding EM. Both the original EM paper and textbooks differ from your derivation.
> > As I suggested, you need to make the discussion more clear and precisely point to a reference that justifies your formulation.

---

> > > ### Author Response · Authors · 2024-11-25
> > >
> > > Thank you for revisiting this point. While our derivation differs in presentation from the original EM paper (Dempster et al., 1977), it aligns with standard interpretations and formulations commonly used in modern ML lecture notes and textbooks. We have provided explanations in the rebuttal **Expectation-Maximization algorithm and the derivation of our method** part, and add additional details of Eq. (5) in the new revision. We reference the following materials that adopt similar derivations:
> > > 1) Probabilistic Machine Learning: Advanced Topics (Kevin P. Murphy, 2023), (Page 284, Eq. 6.139)
> > > 2) https://www.cs.princeton.edu/courses/archive/spr08/cos424/scribe_notes/0311.pdf （Sec. 3）;
> > > 3) https://cs229.stanford.edu/summer2020/CS229_NOTE8.pdf (Page 4-5);
> > >
> > > If there are specific aspects of our derivation you believe are incorrect, we would greatly appreciate further details so that we can clarify further.
> > >
> > > Lastly, we would like to confirm whether our responses have addressed your other concerns. While we understand that different notations or approaches for EM derivations may be unfamiliar, we respectfully suggest that this should not be a sole basis for rejecting the paper.

---

> ### Author Response · Authors · 2024-11-21
> **Official Comment by Authors (Part 2)**
>
> 4. **Performance regarding FastEM and BlindDPS**
>
>     1. Difference from FastEM paper:
>
>         The performance differences stem primarily from the size of the motion blur kernels. Larger kernels pose harder blind inversion tasks, degrading reconstruction quality. Since the FastEM paper does not specify kernel sizes, we tested a range of sizes and found that FastEM only performs well on small kernels but struggles significantly with larger, more complex ones.
>
>         Experimental results of FastEM with both small kernels are included in Appendix F of the revision. We successfully replicated FastEM’s reported performance on small kernels but observed significant performance drops with larger ones in Figure 3 of the original paper.
>
>     2. Difference from BlindDPS paper:
>
>         The original BlindDPS paper reports blind inversion experiments only on the FFHQ dataset, not on ImageNet. FFHQ, which primarily contains human faces, has a simpler and more consistent distribution compared to the diverse and complex data in ImageNet. This difference makes blur kernel estimation significantly easier on FFHQ. As demonstrated in Figure 3 of our paper, we successfully replicated BlindDPS’s strong performance on FFHQ. However, poor blur kernel estimation is observed on ImageNet samples.
>
>         Additional experiments supporting these claims have been included in the updated manuscript (Appendix F). Our comparisons are fair, accurate, and more comprehensive than prior work.
>
> ---
>
> 5. **Additional experiments on GibbsDDRM and Non-Blind Posterior Sampling**
>
>     1. **GibbsDDRM**:
>
>         As suggested, we conducted additional experiments with GibbsDDRM, detailed in the updated manuscript (Appendix F). GibbsDDRM and BlindDPS share a similar framework, both utilizing pixel-space diffusion and dual sampling processes for blind inversion. The primary difference is that GibbsDDRM employs an analytic prior, while BlindDPS uses a score-based prior. Consequently, their reconstruction quality is comparable, as confirmed by our experiments and consistent with the results reported in Table 1 of the original GibbsDDRM paper. These findings further reaffirm that our approach remains superior.
>
>     2. **Non-Blind Posterior Sampling**:
>
>         We computed PSNR metrics for non-blind posterior sampling. It defines an upper bound on reconstruction quality as expected.
>
>
> |          | BlindDPS | FastEM | Ours  | Non-Blind |
> |----------|----------|--------|-------|-----------|
> | ImageNet | 17.31    | 17.36  | 19.35 | 19.42     |
> | FFHQ     | 22.58    | 17.46  | 22.65 | 22.94     |
>
> ---
>
> We thank the reviewer once again for their insightful suggestions. We hope our responses address all concerns and are happy to provide further clarifications in additional discussions. We kindly hope the reviewer could consider raising the score if satisfied with our responses. Thank you!

---

> ### Comment · Reviewer_E53D · 2024-11-27
>
> I thank the authors for modifying the text on EM above Eq 5, which makes it clear now (and seems somewhat different than the original version, e.g. the KL divergence there lacked justification and its second argument was the posterior and not the complete likelihood).
>
> I still argue that the proposed method cannot be termed EM.
>
> Specifically, in the E-step in considered EM approach, replacing q by posterior sampling requires sampling multiple x0 given the current phi, which means full reversed flow akin to the non-blind case (this is obviously computational heavy -- but follows from EM).
> The M-step would require minimizing the likelihood given those x0 samples (after summing their per sample likelihoods).
> Importantly, the EM iterations are associated with the same likelihood p(y;phi).
>
> Your method, on the other hand, seems to diverge significantly from EM.
> There is no full E-step. No full posterior sampling flow given phi, and furthermore, all the iterations that do not enter line 282 in Alg 1 appear to be iterations of prior sampling (no guidance) and not posterior sampling.
> Furthermore, in EM, the iterations are applied to the same likelihood p(y;phi), but in your scheme the likelihood p_t(y;phi) changes at each iteration as the index t changes.
>
> Therefore, the proposed method is at most an EM-inspired technique rather than EM or approximate EM.
>
> To conclude, due to some improvement compared to the first version I sightly increased the score. But I am still concerned about the presentation, the heuristic nature of the method (it does not inherit EM properties and lacks theoretical analysis) and its novelty compared to exiting works.

---

> > ### Author Response · Authors · 2024-11-27
> >
> > Thank you for your detailed and thoughtful feedback. We greatly appreciate your recognition of the clarifications we have made, and we do have addressed typos from the original version. We would like to elaborate on the standard EM algorithm and discuss its connection to our method in more depth.
> >
> > ---
> >
> > **Standard EM**: it alternates between
> >
> > (1) E-step: Sample latent variables from the current estimate of the conditional distribution, $x \sim p_\phi(x|y)$, and compute the expected log-likelihood lower bound $\mathcal{L}(\phi)$.
> >
> > (2) M-step: Maximize $\mathcal{L}(\phi)=E_{x \sim p_\phi (x \mid y)}[\log p_\phi(x)]$ to update parameters $\phi$.
> >
> > This iterative procedure allows the EM algorithm to converge to a local maximum of the observed data log-likelihood, making it a powerful technique for estimation problems involving latent variables.
> >
> > **LatentDEM (Ours)**: we modify the standard E-step of EM for diffusion models, in two aspects
> >
> > (1) Compute the posterior $p_\phi(x|y)$ by learning the score functions $\nabla_x \log p_\phi(x|y)$. Through Bayes rules, we decompose $\nabla_x \log p_\phi(x|y) \to \nabla_x \log p(x)+\nabla_x \log p_\phi(y|x)$, where the prior term $\nabla_x \log p(x)$ is learned by a pre-trained latent diffusion model, the likelihood term has closed-form solution. However, under the diffusion-based framework, it usually takes thousands of timesteps $t \in [0, T]$ to sample, where $x_0$ is the clean sample, $x_t$ is the corresponding noisy sample. Therefore, we have $\nabla_{x_t} \log p_\phi(x_t|y) \to \nabla_{x_t} \log p(x_t)+\nabla_{x_t} \log p_\phi(y|x_t)$. Considering $\nabla_{x_t} \log p_\phi(y|x_t)$ is intractable, we adopt an approximation [1] where $\nabla_{x_t} \log p_\phi(y|x_t) \approx \nabla_{x_0} \log p_\phi(y|x_0)$, which has been demonstrated on various inverse problems.
> >
> > (2) Accelerate the E-step by avoiding full posterior sampling flow. Since we need $\nabla_{x_0} \log p_\phi(y|x_0)$, a common way is to go through the full diffusion sampling process to achieve the $x_0$. However, as we show in Appendix E and Table 6 of the original manuscript, it is extremely time-consuming and does not necessarily lead to better performance. Therefore, we compute $x_0$ from $x_t$ through Tweedie's formula [2], and integrate the EM iteration into each timestep, achieving an efficient and robust solution.
> >
> > ---
> >
> > We hope these detailed derivations can address your following concerns:
> >
> > **Regarding the E-step, we generate posterior samples using diffusion posterior sampling**. For simplicity and robustness, we do not go through the full posterior sampling flow, but approximate the posterior via Tweedie’s formula [2], achieving faster inference and better performance, as shown in lines 1211-1233 and Table 6 from the manuscript. Notably, our approach conducts diffusion posterior sampling rather than simply gradient-based point estimation. Therefore, we would argue that this is our advantage, not a drawback.
> >
> > **Regarding the Alg 1 appearing to be iterations of prior sampling (no guidance)**, we would clarify that we are performing posterior sampling under the guidance of the observation $y$. As we discussed in lines 195-206 in the manuscript, to solve an imaging inverse problem with diffusion priors, we need to replace the unconditional score function with a conditional score function. With Bayesian rules, we have $ \nabla_{x_t} \log p_t(x_t | y) = \nabla_{x_t} \log p_t(x_t) + \nabla_{x_t} \log p_t(y | x_t)$ . That is, achieving posterior sampling by combining the prior sampling $s_\theta(x_t, t)$ and data consistency guidance $\log p(y | \hat{x}_0(x_t))$.
> >
> > **Regarding the same likelihood in EM**, Our $p_t(y;x_t,\phi)$ is an approximation of $p(y;x_0,\phi)$, which is widely used in diffusion-based imaging solvers[1][3] to save inference time, otherwise, it's too computationally expensive.
> >
> > **Regarding the novelty**. We want to emphasize that we are the first to solve blind inverse problems using latent diffusion models, which is non-trivial in robustness and speed. Besides, we conduct comprehensive experiments on a 3D pose-free inverse problem, as no one has done before.
> >
> > ---
> >
> > Once again, we sincerely thank you for your patience and constructive feedback. We hope our response addresses your concerns and highlights the novelty and robustness of our approach, offering a fresh perspective on solving both 2D and 3D blind inverse problems with latent diffusion models.
> >
> > **Reference**
> >
> > [1] Hyungjin Chung, et al. Diffusion Posterior Sampling for General Noisy Inverse Problems, ICLR 2023.
> >
> > [2] Bradley Efron. Tweedies Formula and Selection Bias. 2011.
> >
> > [3] Litu Rout, et al. Solving Linear Inverse Problems Provably via Posterior Sampling with Latent Diffusion Models, NeurIPS 2023.

---

> ### Author Response · Authors · 2024-12-02
> **Anticipation of your response**
>
> Dear Reviewer E53D,
>
> We sincerely thank you for your valuable feedback, time, and efforts. During the rebuttal period, we are happy that most concerns have been addressed, which not only strengthened the paper but also led to positive reevaluation and score improvements from other reviewers. We hope our responses have fully addressed your concerns as well. If not, we are eager to address any remaining concerns, and provide further clarifications or additional experiments as needed. A prompt response would be greatly appreciated to help us make timely improvements. We sincerely look forward to receiving your updated feedback.
>
> Best,
> Authors

---

### Official Review · Reviewer_6ZdJ · 2024-11-04

**Soundness:** 2
**Presentation:** 3
**Contribution:** 2
**Rating:** 5
**Confidence:** 3

**Summary:**

This paper aims to tackle the challenging problem of blind inverse tasks by employing latent diffusion priors within an expectation-maximization (EM) framework, which iteratively estimates both the unknown measurement operator (within M-step) and the underlying image (within E-step). The proposed approach has been validated for tasks of 2D blind deburring and 3D pose-free sparse-view reconstruction. The parameters of the forward model are estimated in the M-step through using a pre-trained denoiser of Gaussian noise within a PnP framework. In the E-step, a so called "annealing consistency" approach is proposed through introducing a tunable hyper-parameter in the score of likelihood term to stabilize the training. In addition to the score of likelihood term,  a "gluing" regularization (Rout et al, 2024) is employed within diffusion posterior sampling for 2D deblurring task.

**Strengths:**

This problem holds broad interest across the machine learning and signal processing communities.
The paper is generally well-structured and somewhat clear.

**Weaknesses:**

Several technical ambiguities require clarification. These include the choice of likelihood formulation, the role of “gluing” regularization versus annealing consistency, and the performance discrepancies across datasets. More experimental validation, particularly isolating the effects of different regularization techniques, would strengthen the claims. Additionally, a detailed comparison of hyperparameters with competing methods would improve transparency and reproducibility. For further information of the detected ambiguities please refer to the section "Questions".

**Questions:**

the following issues and ambiguities require further clarification or validation:

1. While (Chung et al. 2022b) formulates the score of the data likelihood as $\nabla_{z_t} \log p_t(y|z_t)$ in Eq.(14), this paper opts for Eq.(9), citing “annealing consistency” for enhanced stability and performance. The authors should provide empirical evidence or theoretical justification for this choice. How does “annealing consistency” compare to the “vanilla” latent DPS term? Also, clarify the behavior of Eq.(9) and Eq.(18) when measurements are noiseless, i.e.  $\sigma = 0$.

2. For the 2D blind deblurring task, Eq.(16) includes a “gluing” regularization term (in addition to “vanilla” latent DPS-based term) as per (Rout et al. 2024). It is unclear whether the reported improvements are due to this regularization or the proposed “annealing consistency.” An experiment isolating these factors in 2D deblurring would be helpful.

3. The paper should explain why BlindDPS performs much better on FFHQ than ImageNet, and detail the training setup. Additionally, comparing FID metrics between this work, FastEM (Rombach et al. 2022), and BlindDPS (Chung et al. 2022a) would improve clarity [as used in the aforementioned papers as well]. Why does FastEM show negligible improvement over initial observations (e.g., see the estimated blurring kernel which is close to Dirac delta)?

4. Clarify the parameter $\phi$ for widely studied image inverse tasks (e.g., compressed sensing, tomography, super-resolution, inpainting, spectral imaging) to enhance understanding of its role across applications.

5. The paper aims to tackle [general] ‘blind’ inverse problems through estimating the parameters of forward measurement operator in the M-step of the EM framework as eq.(10).  To this aim, the parameter $\phi$ (e.g., blurring kernel for the task of 2D deblurring) is estimated through a PnP denoising step as eq.(19). Please elucidate on whether a denoiser pre-trained on Gaussian noise can generalize to other inverse tasks like compressed sensing, and explain what signal regularities it might exploit in estimating $\phi$.

6. A comprehensive list of trainable and tunable hyperparameters for the proposed method, along with those used in Fast Diffuse EM (Laroche et al. 2024) and BlindDPS (Chung et al. 2022a), would facilitate fair comparisons.

7. Please specify the network used in the E-step and provide experimental setup details for Table 1.

Minor Comments:

- In the context “these priors cannot capture the complex natural image distributions, limiting the solvers’ ability to produce high-quality reconstructions”, please consider replacing the (Candes & Romberg, 2007) citation with the following, which better aligns with the stated limitations of image priors:
https://doi.org/10.1109/TIP.2005.863057
https://doi.org/10.1109/ICIP.2007.4379013
https://doi.org/10.1109/GlobalSIP.2013.6737048
https://doi.org/10.1137/16M1102884

- Please clarify initialization of $\mathcal{A}_{\phi^{(t)}}$ in Algorithm 1 and specify corresponding M-step equations (perhaps Appendix C.2 ?)

- Differences between Eq.(11) of this paper and Eq.(24)/Eq.(B.9) in (Laroche et al. 2024) require clarification.

- Please correct discrepancies in Section 4.3 regarding the noiseless assumption in Eq.(1).

- Minor typographical and grammatical errors include:
  * Typo corrections: “a intermediate” and “an one-to-many.”
  * Redundant citations: “Liu et al. (2023b)” and “Liu et al. (2023c)” refer to the same work.
  * Misuse of “While” and “However” following Eq.(4).
  * Replace “introduce” with “utilize” or “employ” when referring to “gluing” regularization as it is not part of the contribution of this work.
  * The paper overlooks the following relevant work:
        Murata et al., “Gibbsddrm: A partially collapsed Gibbs sampler for solving blind inverse problems with denoising diffusion restoration.” ICML, 2023.

---

> ### Author Response · Authors · 2024-11-21
> **Official Comment by Authors (Part 1)**
>
> We thank the reviewer for their constructive suggestions and thoughtful feedback. We are pleased that the reviewer recognizes the broad interest of our research problem across the machine learning and signal processing communities. Below, we provide detailed explanations and clarifications for the concerns raised:
>
> ---
>
> 1. **Annealing Consistency and the noiseless condition**
>
> Empirical evidence and theoretical analysis of the annealing consistency technique are presented in Figure 5 and Appendix B of the original paper. This technique significantly enhances imaging quality by adaptively balancing the weights between data fidelity and generative priors based on model accuracy.
> Regarding the noiseless condition ($\sigma=0$), we use a small virtual uncertainty (e.g., $\sigma=1e-2$) to ensure sampling stability. This approach is consistent with prior work, including the classical DPS and LatentDPS algorithms.
>
> ---
>
> 2. **Role of “Gluing” regularization vs. Annealing Consistency**
>
> The effectiveness of the proposed annealing consistency technique is experimentally demonstrated in Figure 5 of the original paper. The second column (gluing without annealing) is compared to the third column (both gluing and annealing), showing that annealing consistency clearly improves performance.
> We agree that an isolation study would better quantify the individual contributions of “gluing” regularization and “annealing consistency.” According to your suggestion, we included additional experiments in the updated manuscript (Appendix F), removing the gluing term while retaining annealing consistency. Results show that both techniques independently improve reconstruction, but their combination achieves the best performance.
>
> ---
>
> 3. **More discussions on baseline performance**
>
>     1) **Performance Difference on FFHQ vs. ImageNet**: The FFHQ dataset, dominated by human faces, has a simpler and more consistent distribution compared to the diverse and complex ImageNet dataset. This complexity makes it harder for BlindDPS to estimate motion blur kernels on ImageNet, consequently influencing the imaging quality. Notably, BlindDPS only reports results on FFHQ in their original paper, likely due to these challenges.
>
>     2) **FID Metric Comparison**: As requested, we computed the FID score for our method and compared it with FastEM and BlindDPS (below table). The results confirm our method's superiority.
>
>     | | **FastEM** | **BlindDPS** | **Ours** |
>     |---------------------------|------------|--------------|----------|
>     | FID Score (ImageNet)  | 58.91      | 53.35        | 49.21    |
>
>
>     3) **Performance of FastEM**: The original FastEM paper does not specify the size of the blurring kernels used. To address this, we tested FastEM across various settings. Our experiments reveal that FastEM is unstable when dealing with large or complex kernels and does not improve a lot from the initial observations, as the reviewer pointed out. The results reported in the original FastEM paper primarily focus on simple kernels, whereas our experiments use larger kernels, which is consistent with BlindDPS’s settings. Additional results with small kernels are included in the revised manuscript (Appendix F, Fig. 15), and these findings do not influence our conclusions.
>
> ---
>
> 4. **Parameter $\phi$ Across Applications**
>
>  Parameter $\phi$ represents any parameter that defines the forward model, such as blurring kernels in deblurring, masks in inpainting, frequency masks in compressed sensing, or scanning angles in computed tomography. We will add further explanations in the revised manuscript.
>
> ---
>
> 5. **Generalization of the PnP Denoiser**
>
> The PnP denoiser trained on Gaussian noise can serve as a general regularizer for inverse imaging problems, as theoretically and experimentally proved in prior works [1, 2]. Grounded in the theory of denoising score matching [3], Gaussian denoisers capture a wide range of image priors, from low-level properties (e.g., sparsity, smoothness) to high-level features (e.g., semantic and structural coherence). Due to the page limit, we could not provide detailed explanations in the original paper. We are happy to include them in the revised version.
>
> ---
>
> 6. **Comprehensive list of hyperparameters**
>
>  We have added a comprehensive table of tunable hyperparameters for our method, as well as those used by FastEM and BlindDPS, in the updated manuscript. Key differences are summarized below:
>
> | Method    | Steps | Noise Scheduler | Model Prediction | M-step             | Lambda | Gamma |
> |-----------|-------|-----------------|------------------|--------------------|--------|-------|
> | FastEM    | 1000  | linear          | Epsilon          | 10 HQS iterations | 1      | 1e5   |
> | BlindDPS  | 1000  | linear          | Epsilon          | 1 diffusion process| 1      | /     |
> | Ours      | 1000  | linear          | Epsilon          | 10 HQS iterations | 1      | 5e6   |

---

> ### Author Response · Authors · 2024-11-21
> **Official Comment by Authors (Part 2)**
>
> 7. **Network architecture and experimental setup for Table 1**
>
> The E-step used the Stable Diffusion v-1.5 model, as described in Sec. 5.1 of the original paper. The M-step employs a DnCNN model with 5 blocks and 32 channels[4], as explained in Appendix C.2.
>
> Additional experimental details for Table 1 are included in Appendix C, and we followed official implementations for BlindDPS and FastEM. We will add further clarifications to the revised manuscript.
>
> ---
>
> 8. **Other minor points**
>
> We have updated citations, initialization details, and typographical errors as per your suggestions.
> There is no difference between Equation (11) in our paper and Equation (24)/Equation (B.9) in FastEM.
> Additionally, we have incorporated GibbsDDRM [5] into the literature review and performed additional experiments using it as a new baseline (Appendix F).
>
> We thank the reviewer once again for their insightful suggestions. We hope our responses address all concerns and are happy to provide further clarifications in additional discussions. Thank you!
>
> ---
>
> **Reference**
>
> [1] S. V. Venkatakrishnan, et al. Plug-and-play priors for model-based reconstruction. IEEE GlobalSIP, 2013.
>
> [2] Yu Sun, et al. An online plug-and-play algorithm for regularized image reconstruction. IEEE TCI, 2019.
>
> [3] Vincent P. A connection between score matching and denoising autoencoders. Neural computation, 2011.
>
> [4] Kai Zhang, et al. Plug-and-play Image Restoration With Deep Denoiser Prior, TPAMI 2021.
>
> [5] Murata, et al. Gibbsddrm: A partially collapsed Gibbs sampler for solving blind inverse problems with denoising diffusion restoration. ICML, 2023.

---

> > ### Comment · Reviewer_6ZdJ · 2024-11-27
> > **Rebuttal Review, Part 1**
> >
> > Thank you for the responses, which addressed some of my concerns. However, I remain unconvinced by certain answers, and significant issues persist.
> > ---------------------------------------------------------------------------------------------------
> >
> > 1) Annealing Consistency: a heuristic and non-novel contribution
> > -------------
> > The proposed “annealing consistency” is heuristic and not an original contribution of this paper. Appendix B cites Chung et al. (2022b), where a similar approach was proposed in 'score ALD' (https://arxiv.org/abs/2108.01368), using annealing in the likelihood score (see Eq.(4) of 'score ALD'). Such annealing is also not new; for example, see https://doi.org/10.1109/TIP.2018.2875569.
> >
> > Contrary to the authors’ claims, Appendix B provides no 'theoretical explanation' for “annealing consistency” beyond possibly suggesting replacing the denominator in Eq.(14) with that in Eq.(9). Additionally, the regularization parameter in “annealing consistency” is not novel. Several earlier diffusion-based works include similar regularization terms between the noisy data score and the likelihood score or guidance term (Eq.(6) of the original draft). Examples include:
> > - P. Dhariwal et al., “Diffusion Models Beat GANs on Image Synthesis,” NeurIPS 2021 (see Section 4, and Eq.(4.1) in H. Chung, “Review of Diffusion Model Theory and Applications,” J. Korean Soc. Ind. Appl. Math, 2024).
> > - J. Song et al., “Pseudoinverse-Guided Diffusion Models for Inverse Problems,” ICLR 2023 (see Eq.(4) and Eq.(8)).
> > - H. Chung et al., “Diffusion Posterior Sampling for General Noisy Inverse Problems,” ICLR 2023 (see Algorithms 1 and 2, and the text after Eq.(7) in X. Peng et al., “Improving Diffusion Models for Inverse Problems Using Optimal Posterior Covariance,” ICML 2024).
> >
> >
> >
> > 2) the performance/role of “gluing” vs. annealing consistency is unclear
> > -------------
> > Regarding the role of “gluing” vs. annealing consistency, the authors’ response is insufficient. Figure 14 in the revised draft shows one example where gluing+annealing outperforms their individual counterparts, but this is incomplete, as a generative method can produce varied-quality samples. My request for “an experiment isolating these factors in 2D deblurring” referred to a study comparable to Table 1. Moreover, contrary to the authors’ response, neither Figure 5 nor its caption/text clarifies that “No Annealing” refers to “gluing,” leaving this unclear to readers. Besides, the proposed work seems to be just combination of already existing tools/methods developed for the same task.

---

> > > ### Comment · Reviewer_6ZdJ · 2024-11-27
> > > **Rebuttal Review, Part 2**
> > >
> > > 3) Estimation of $\phi$ and generalizability of the proposed method:
> > > -------------
> > > The authors’ response is inadequate.
> > >
> > > For the examples cited (e.g., compressed sensing or inpainting), the forward model parameters can be directly derived from the measurement vector $y$—e.g., the frequency mask from the Fourier plane in compressed sensing or the inpainting mask from the image domain. However, for general blind inverse problems, it remains unclear how the unknown parameter $\phi$ of the forward operator $\mathcal{A}_{\phi}$ is estimated in the M-step of the EM framework (Eq.(10)) using the PnP framework.
> > > The authors claim, “The PnP denoiser trained on Gaussian noise can serve as a general regularizer for inverse imaging problems.” While true, this is because natural signals exhibit complex regularities captured by sophisticated denoisers as implicit prior models rather than simple proximal operators. For tasks like deblurring, estimating a binary blur kernel using PnP (e.g., FastDiffusionEM by Laroche et al., 2024) may be feasible. However, identifying regularities within $\phi$ for general inverse problems is neither realistic nor practical.
> > >
> > >
> > > 4) Overlooked work and biased comparison
> > > -------------
> > > The paper overlooks a key related work, GibbsDDRM:
> > >
> > > Murata et al., “Gibbsddrm: A partially collapsed Gibbs sampler for solving blind inverse problems with denoising diffusion restoration.” ICML, 2023.
> > >
> > > Instead, the authors present biased experimental results in Figure 16 of the appendix, failing to discuss this prior work, its similarities, or its differences with the current study. They claim GibbsDDRM performs significantly worse than the proposed method but provide no justification for the discrepancy. For instance, why do the results on the FFHQ dataset in Figure 16 differ markedly from those in the GibbsDDRM paper (e.g., Figures 1, 5, 6, 8, and 9)? Are the experimental settings (e.g., kernel size, kernel type, noise standard deviation) consistent with those in GibbsDDRM?
> > >
> > > 5) Still, some of minor comments are overlooked.
> > > -------------
> > > I believe citing (Candes & Romberg 2007) is out of context. “gluing” is not the introduced in this work but rather ‘employed’. The initialization of $\mathcal{A}_{\phi}$ is unknown.
> > >
> > > -------------
> > > ---------------------------------------------------------------------------------------------------
> > >
> > > Considering the above concerns, I am unable to raise my evaluation score.
> > > ---------------------------------------------------------------------------------------------------

---

> > > > ### Author Response · Authors · 2024-11-29
> > > >
> > > > **Regarding the estimation of $\phi$ and generalizability of the proposed method**: We agree that the plug-and-play denoiser works better for natural signals with complex regularities, such as kernels in image deblurring, but it may not be the best choice for all inverse problems. In fact, for the 3D imaging task in our paper, we did not use a denoiser to regularize the underlying poses. Our primary focus is on the Diffusion EM approach for blind inversion, and the denoiser is just one possible technique for parameter estimation when needed. We will add more clarifications in our final version.
> > > >
> > > > **Regarding GibbsDDRM**: Additional experiments with GibbsDDRM are in Appendix F. We are pretty sure that the experimental settings between GibbsDDRM and ours are identical (both use $64 \times 64$ motion blur kernels, along with the same measurement noise). However, we cannot get good reconstruction results if we simply run their official implementation and checkpoint on the validation data. Today we have tried our best to tune the hyper-parameters of the GibbsDDRM algorithms, including the normalization scale, step size of updating the kernel and weighting of the Laplace prior to achieving better results, as reported in Figure 16 of the revision. These parameters do influence the final results, and the best parameter for different images seems not exactly the same. We are committed to providing the best results in the final revision.
> > > >
> > > > We are also happy to add more discussion about GibbsDDRM in the related work section in our final submission (sorry today is the last day for updating the PDF and we didn't finish this task before deadline.) GibbsDDRM and BlindDPS, one baseline for both GibbsDDRM and our method, share a similar framework, both utilizing pixel-space diffusion and dual sampling processes for blind inversion. The primary difference is that GibbsDDRM employs an analytic prior (Laplace prior), while BlindDPS uses a score-based prior. Their reconstruction quality is comparable, as reported in Table 1 of the original GibbsDDRM paper and confirmed by our experiments, while GibbsDDRM involves more tunable hyperparameters. They are both different from our work in image priors (Latent Diffusion v.s. Pixel-based Diffusion) and sampling techniques (EM v.s. Dual Diffusion).
> > > >
> > > >
> > > > **Regarding minor comments**, we have removed the citation of (Candes & Romberg 2007), and replaced 'introduce' with 'employ' in the revision. The initialization of the blur kernel $\mathcal{A}_\phi$ follows the same procedure as in FastEM. Specifically, we initialize the $64 \times 64$ motion blur kernel using a $64 \times 64$ Gaussian blur, where only a few pixels in the center are randomly set to 1.
> > > >
> > > > ---
> > > >
> > > > Once again, we thank you for your constructive and detailed comments. We hope these additional clarifications will strengthen the manuscript and address your remaining concerns.
> > > >
> > > > **Reference**
> > > >
> > > > [1] P. Dhariwal et al., “Diffusion Models Beat GANs on Image Synthesis,” NeurIPS 2021.
> > > >
> > > > [2] H. Chung, “Review of Diffusion Model Theory and Applications,” J. Korean Soc. Ind. Appl. Math, 2024.
> > > >
> > > > [3] J. Song et al., “Pseudoinverse-Guided Diffusion Models for Inverse Problems,” ICLR 2023.
> > > >
> > > > [4] H. Chung et al., “Diffusion Posterior Sampling for General Noisy Inverse Problems,” ICLR 2023
> > > >
> > > > [5] X. Peng et al., “Improving Diffusion Models for Inverse Problems Using Optimal Posterior Covariance,” ICML 2024.
> > > >
> > > > [6] Murata et al., “Gibbsddrm: A partially collapsed Gibbs sampler for solving blind inverse problems with denoising diffusion restoration.” ICML, 2023.

---

> > > > > ### Comment · Reviewer_6ZdJ · 2024-12-03
> > > > >
> > > > > Regarding the estimation of $\Phi$ and the generalizability of the proposed method, your response remains unconvincing. As I stated earlier, there is no compelling evidence provided for why a Gaussian denoiser should generalize across all forward models to estimate their parameters.
> > > > >
> > > > > I appreciate the clarification on the poor performance of GibbsDDRM and hope its authors can resolve the issue or provide an explanation based on their code.

---

> > > ### Author Response · Authors · 2024-11-29
> > >
> > > Thank you for your thoughtful feedback and for raising important concerns. Below, we address the concerns in detail:
> > >
> > > ---
> > >
> > > **Regarding the “annealing consistency” approach**, we agree that annealing has been preliminarily explored in diffusion-based image reconstruction[1][2][3][4][5].
> > > Specifically, [1][2] introduce scaling factor $s$ and $\omega$ respectively to scale classifier gradients, balancing the quality and diversity of the generated samples; [3] introduces $\gamma_t=\sigma_t^2 / (\sigma_t^2+1)$ to handle measurement noise; [4][5] modify the step size $\zeta$ and $\delta$ according to the data consistency empirically for better performance.
> > > However, our annealing technique has an intrinsic physical meaning, as explained in Appendix B,  we use it to mitigate forward model error in the context of blind inversion. Our goal is not to introduce the concept of annealing itself, but rather to solve the blind inverse problems based on latent diffusion models, where the sampling process can be unstable due to the unknown parameters of the forward model. Therefore, tailoring the annealing technique for blind inversion significantly reduces the negative effects of the inaccurate forward model in the early EM steps.
> > > The annealing technique is a must for our task, as demonstrated by the qualitative results in Figure 14 of the revision and the quantitative results below.  We will revise the manuscript to better clarify the novelty of our approach in relation to prior work.
> > >
> > > |                         |          |       |       |        |       |       |       |       |        |       |
> > > |-------------------------|----------|-------|-------|--------|-------|-------|-------|-------|--------|-------|
> > > | Method                  | ImageNet |       |       |        |       | FFHQ  |       |       |        |       |
> > > |                         | Image    |       |       | Kernel |       | Image |       |       | Kernel |       |
> > > |                         | PSNR  | SSIM  | LPIPS | MSE    | MNC   | PSNR  | SSIM  | LPIPS | MSE    | MNC   |
> > > | Vanilla                 | 18.88   | 0.428  |0.349   | 0.013    |0.404   | 21.40   | 0.603   | 0.261   | 0.011    | 0.414   |
> > > | Gluing                  | 18.94   | 0.440   | 0.3289   | 0.011    |0.421   | 21.81   |0.610  | 0.211   |0.011    | 0.428   |
> > > | Annealing               | 19.06   |0.457   | 0.2742   | 0.011    | 0.434   | 22.58   | 0.631   | 0.200   | 0.010    | 0.430   |
> > > | Gluing+Annealing (Ours) | 19.35    | 0.496 | 0.256 | 0.010   | 0.441 | 22.65 | 0.653 | 0.167 | 0.009  | 0.459 |
> > > |                         |          |       |       |        |       |       |       |       |        |       |
> > >
> > >
> > > **Regarding the performance/role of “gluing” vs. annealing consistency is unclear**, we have conducted a full study that is comparable to Table 1. The results are shown in Figure 14 of the revision and the quantitative results above. Additionally, we further clarified that “No Annealing” refers to “gluing” in the caption of Figure 5 as suggested.
> > >
> > > **Reference**
> > >
> > > [1] P. Dhariwal et al., “Diffusion Models Beat GANs on Image Synthesis,” NeurIPS 2021.
> > >
> > > [2] H. Chung, “Review of Diffusion Model Theory and Applications,” J. Korean Soc. Ind. Appl. Math, 2024.
> > >
> > > [3] J. Song et al., “Pseudoinverse-Guided Diffusion Models for Inverse Problems,” ICLR 2023.
> > >
> > > [4] H. Chung et al., “Diffusion Posterior Sampling for General Noisy Inverse Problems,” ICLR 2023
> > >
> > > [5] X. Peng et al., “Improving Diffusion Models for Inverse Problems Using Optimal Posterior Covariance,” ICML 2024.
> > >
> > > [6] Murata et al., “Gibbsddrm: A partially collapsed Gibbs sampler for solving blind inverse problems with denoising diffusion restoration.” ICML, 2023.

---

> > > > ### Comment · Reviewer_6ZdJ · 2024-12-03
> > > >
> > > > Thank you for addressing some ambiguities, such as the performance comparison between annealing and “gluing” through the provided results table. I also appreciate the authors' acknowledgment that annealing is not a novel idea.
> > > >
> > > > However, your claim that “Our annealing approach is … to mitigate forward model error,” while other methods aim to “handle measurement noise,” is inaccurate. The goal of approaches like score ALD (Dhariwal et al., NeurIPS 2021), IDBP (Tirer & Giryes, IEEE TIP 2019), $\pi$GDM (Song et al., ICLR 2023), and DPS (Chung et al., ICLR 2023) is to either regularize the likelihood score $\nabla_{z_t} \log p_t(y|z_t)$ or enforce the consistency of $\hat{z}_0$ with the measurements $y$.
> > > >
> > > > Additionally, the claim that “previous methods empirically modify the step size according to data consistency, whereas our annealing has intrinsic physical meaning” is not supported. For instance, $\pi$GDM uses similar reasoning (see Eq.(4) of Song et al., ICLR 2023, where $p_t(z_0|z_t) \approx \mathcal{N}(\hat{z}_0, \nu_t^2 I)$) to derive its regularization terms. Both $\pi$GDM and your work heuristically select the regularization parameter. Thus, “annealing consistency” is not novel but rather an adaptation of earlier approaches that also happens to work here.

---

> ### Author Response · Authors · 2024-12-02
> **Anticipation of your response**
>
> Dear Reviewer 6ZdJ,
>
> We sincerely thank you for your valuable feedback, time, and efforts. During the rebuttal period, we are happy that most concerns have been addressed, which not only strengthened the paper but also led to positive reevaluation and score improvements from other reviewers. We hope our responses have fully addressed your concerns as well. If not, we are eager to address any remaining concerns, and provide further clarifications or additional experiments as needed. A prompt response would be greatly appreciated to help us make timely improvements. We sincerely look forward to receiving your updated feedback.
>
> Best,
> Authors

---

> > ### Comment · Reviewer_6ZdJ · 2024-12-02
> > **My final review will be submitted shortly**
> >
> > Dear Mauricio,
> >
> > Thank you for you detailed response to my concerns and questions.
> > My final review will be submitted shortly.
> >
> > BR,
> > Reviewer 6ZdJ

---

> > ### Comment · Reviewer_6ZdJ · 2024-12-03
> > **Final decision**
> >
> > I understand and value the practical relevance of your empirical contributions for blind inverse problems. However, the lack of theoretical innovation makes the paper less compelling in my view. I have kept my confidence score low to allow other reviewers and the AC, who may better assess the empirical contributions, to decide on the paper's merit for publication at ICLR.

---

> > > ### Author Response · Authors · 2024-12-04
> > >
> > > We thank the reviewer for the valuable feedback. We are happy that most concerns have been addressed. Below, we provide clarifications for the remaining two points.
> > >
> > > **Regarding annealing**:  While we acknowledge that other methods also employ annealing techniques to regularize the likelihood score or enforce data consistency, these approaches primarily aim to mitigate the **approximation error** introduced by theoretical derivations, such as Tweedie's formula. These approximations are not necessary for more advanced diffusion posterior sampling approaches, such as plug-and-play Monte Carlo (PMC)[1]. In contrast, our proposed annealing technique is specifically designed for blind inversion tasks, where it has intrinsic physical significance in addressing **physical model error**, such as inaccuracies in blur kernels or camera poses during the early EM iterations. Although annealing is not a pure innovation in optimization and inverse problems, we would like to emphasize that the design philosophy behind our approach is fundamentally different, and other annealing techniques are not tailored to this specific scenario.  We will further clarify it in the revision.
> > >
> > >
> > > **Regarding Gaussian denoiser**: We would like to emphasize again that we didn't say Gaussian denoiser is the best and should be used for all imaging inverse problems - indeed, we didn't use it in our 3D tasks.  However, we would like to offer further insight into why a Gaussian denoiser can be a reasonable regularizer for most imaging problems. If a model can generate a specific type of signal, it implicitly defines their distribution and can be used as a regularizer [2][3][4]. General Gaussian denoisers have been proven to be an efficient tool for learning generative models [5], and more specifically, diffusion models, which can approximate a wide range of continuous parameters. Given this, we didn't see why this would become an important factor for underming the basis of our proposed algorithm.
> > >
> > >
> > > We hope these additional clarifications can address all remaining concerns and will incorporate the above discussion in the revision to make it clear.  Once again, we thank the reviewer for the constructive comments!
> > >
> > >
> > > **Reference**
> > >
> > > [1] Yu Sun, et al. Provable Probabilistic Imaging Using Score-Based Generative Priors, IEEE TCI 2024.
> > >
> > > [2] SV Venkatakrishnan, et al. Plug-and-Play priors for Model based Reconstruction, IEEE Global Conference on Signal and Information Processing, 2013.
> > >
> > > [3] R Ahmad, et al. Plug-and-Play Methods for Magnetic Resonance Imaging: Using Denoisers for Image Recovery, IEEE Signal Processing Magazine, 2020.
> > >
> > > [4] Y Romano, et al. The little engine that could: Regularization by denoising (RED), SIAM Journal on Imaging Sciences, 2017.
> > >
> > > [5] Jonathan Ho, et al. Denoising diffusion probabilistic models, NeurIPS 2020.

---

> ### Author Response · Authors · 2024-12-03
>
> Thanks for your kind response! We look forward to your final evaluation.

---

### Author Response · Authors · 2024-11-21

We sincerely thank the reviewers for their constructive and detailed feedback. Below, we summarize the major concerns and our responses. Point-by-point replies have been provided to each reviewer, and the manuscript has been updated accordingly.

---

1. **Contribution and Novelty**

    1.  Pre-trained LDMs for Blind Inverse Problems:

        This work is the first to use pre-trained latent diffusion models (LDMs) for challenging blind inverse problems. The integration of LDMs and EM framework is non-trivial because of two major issues:

        (a) Instability: The encoder-decoder structure of LDMs, combined with unknown model errors, makes diffusion posterior sampling fragile.

        (b) limited efficiency: iterative gradient backpropagation through the decoder leads to substantial computational costs.

        To address these issues, we introduced **annealing consistency** and **skip-gradient techniques** that effectively stabilize and accelerate the inference process, enabling our method to outperform existing approaches, such as FastEM and BlindDPS.

    2. New Applications:

        We demonstrate a novel application of our method in **pose-free sparse-view 3D reconstruction**, a high-dimensional, non-linear inverse problem more challenging than blind deblurring. This success further highlights the robustness and versatility of our algorithm, providing a new benchmark for testing.

---

2. **Mathematical Formulation of the EM Framework**

Our approach is firmly grounded in the EM algorithm, not an alternating optimization scheme. In our method:

1) The **E-step** involves posterior sampling from the diffusion prior.

2) The **M-step** maximizes the expected log data fidelity to estimate model parameters.

The only approximation lies in using Tweedie’s formula in the E-step to accelerate computation, which does not alter the EM framework's fundamental nature. While the EM derivation in the original paper is derived from a variational Bayes perspective, we have provided additional mathematical derivations following the classic algorithm, and added more explanations and discussions in our detailed responses to each reviewer.

---

3. **Fair Comparison with Baselines**

The differences in reconstruction scores between our work and prior methods primarily arise from variations in kernel sizes and datasets: The original FastEM paper tested only on small kernels, and the original BlindDPS paper evaluated only on FFHQ and AFFHQ datasets. In contrast, our experiments cover a broader range of kernel sizes, complexities, and datasets (e.g., ImageNet). Additional experiments, included in the updated manuscript and reviewer responses, confirm that our comparisons are **fair, accurate, and more comprehensive** than previous work.

---

We hope these clarifications and additional results address the reviewers' concerns. We are happy to discuss any remaining points during the discussion phase. Thank you again for your insightful feedback.

---

### Author Response · Authors · 2024-12-02
**Rebuttal Summary**

### **Dear Reviewers, ACs, and PCs,**

We sincerely thank all the reviewers for their valuable and constructive feedback, as well as for dedicating their time to reviewing our paper. Based on the insightful suggestions provided during the rebuttal phase, we have conducted a thorough revision addressing the reviewers' key concerns. Below, we summarize these concerns and detail the revisions and updates included in the final submission. We hope this provides the reviewers, ACs, and PCs with a clearer understanding of the progress and outcomes of the rebuttal discussion.

---

### **Concerns**

+ **[6ZdJ, E53D, rjXb]** Expectation-Maximization algorithm and the derivation of our method.
  + **[Authors]**  We explain in the response. *[Response to 6ZdJ, E53D, rjXb] [General Response: Mathematical Formulation]*
  + **[Authors]** Additional clarifications and step-by-step derivation . *[L205-L223] [Appendix G]*
+ **[6ZdJ, E53D, yUwp]** Comparison with baselines.
  + **[Authors]**  We explain in the response. *[Response to 6ZdJ, E53D, yUwp] [General Response: Fair Comparison with Baselines]*
  + **[Authors]** Additional experiments. *[L1409-1426, Figure 15]*
+ **[6ZdJ, E53D, rjXb]** Detailed experimental settings.
  + **[Authors]** We explain in the response. *[Response to 6ZdJ, E53D, rjXb]*
+ **[E53D, yUwpn]** Contribution and novelty.
  + **[Authors]** We explain in the response. *[Response to E53D, yUwpQ] [General Response: Contribution and Novelty]*
+ **[6ZdJ, E53D]** Missing reference of GibbsDDRM.
  + **[Authors]** We explain in the response. *[Response to 6ZdJ, E53D]*
  + **[Authors]** We add discussion and experiments. *[L1438-L1452, Figure 16]*
+ **[6ZdJ]** Role of annealing consistency technique.
  + **[Authors]**  We explain in the response. *[Response to 6ZdJ]*
  + **[Authors]** We conduct additional isolating experiments to demonstrate the proposed technique. *[L1350-L1377, Figure 14]*

---

### **Revisions**

+ **[Preliminary]** We restate the derivation of the Expectation-Maximum algorithm to make it precise, to address concerns raised by Reviewer 6ZdJ, E53D and rjXb. *[L 205-233, Preliminary]*
+ **[Method]** We rewrite the Half-Quadratic Splitting (HQS) optimization according to Reviewer rjXb's concern. *[L332-L337, Section 4.3]*
+ **[Appendix F]** We add all additional experimental results here for clarification. *[L1382-1493, Appendix F]*
+ **[Appendix G]** We derive the Expectation-Maximum algorithm step-by-step, according to concerns raised by Reviewer 6ZdJ, E53D and rjXb. *[L1514-1565, Appendix G]*

---

### **Other Points**

+ **[6ZdJ]** Replace “introduce” with “employ”. *[L 853, Appendix C.1]*
+ **[6ZdJ]** Typos. *[Corrected]*
+ **[rjXb, 6ZdJ]** Citations miused. *[Corrected]*

---

### **Summary**

We sincerely thank the reviewers for their valuable suggestions, which have helped us strengthen our revised submission. We believe the revised submission could be much more robust and contribute to broadening the field of diffusion-based inverse imaging, offering valuable insights to the community. Notably, our approach not only excels at 2D blind deblurring, but also achieves superior performance on the challenging pose-free multi-view consistent 3D reconstruction. We deeply appreciate the efforts of the reviewers, ACs, and PCs during the rebuttal period.

*Best regards,*

*Authors*

---

### Meta-Review · Area_Chair_bEn3 · 2024-12-16

**Metareview:**

This paper proposes LatentDEM, a novel method for addressing blind inverse problems through latent diffusion models within an Expectation-Maximization (EM) framework. The authors claim that their method outperforms existing methods for 2D blind deblurring and enables new capabilities in non-linear 3D inverse rendering problems, such as pose-free sparse-view 3D reconstruction.

Strengths:
- The paper is generally well-written and easy to follow.
- It proposes a novel method for addressing blind inverse problems using latent diffusion models.
- The method is shown to be effective for both 2D and 3D inverse problems.

Weaknesses:
- Some reviewers have raised concerns about the clarity and accuracy of the EM formulation.
- There are ambiguities regarding the likelihood formulation, the role of "gluing" regularization versus annealing consistency, and the handling of performance discrepancies across datasets.
- The novelty of the "annealing consistency" technique is questioned.
- More experimental validation is needed, particularly isolating the effects of different regularization techniques and providing a detailed hyperparameter comparison for reproducibility.

The decision to reject this paper is based on the following reasons:
- The concerns about the EM formulation are not fully addressed. This hinders the understanding of the proposed formulation.
- The technical ambiguities and the lack of novelty of the "annealing consistency" technique raise concerns about the overall soundness of the method.
- The experimental validation is not sufficient to support the claims.

This paper presents interesting ideas but their potential impact remains unclear due to the current presentation.  A significant revision is necessary to address the reviewers' concerns and fully articulate the paper's contributions. In particular, the following aspects could help make the work clearer and better:

- Clearly respond to concerns regarding the E and M steps.
- Discuss the drawbacks of the proposed methods transparently.
- Clarify that annealing consistency is an existing technique, not a novelty of the paper.
- Provide a comparison between annealing and gluing performances.
- Review, cite, and compare overlooked related works.
- Detail the experimental setup and configurations for all methods.
- Elaborate on the poor performance of comparative methods.

**Additional Comments On Reviewer Discussion:**

During the rebuttal period, the authors have addressed some of the reviewers' concerns, such as providing more details about the experimental setup and conducting additional experiments to isolate the effects of different regularization techniques. However, they have not fully addressed the concerns about the EM formulation and the novelty of the "annealing consistency" technique.

---

### Decision · Program_Chairs · 2025-01-22

Reject